# Change Point Detection via Multivariate Singular Spectrum Analysis

**Arwa Alanqary**
Computational Science and Engineering
MIT
alanqary@mit.edu

**Abdullah Alomar**
EECS
MIT
aalomar@mit.edu

**Devavrat Shah**
EECS
MIT
devavrat@mit.edu

## Abstract

The objective of change point detection (CPD) is to detect significant and abrupt changes in the dynamics of the underlying system of interest through multivariate time series observations. In this work, we develop and analyze an algorithm for CPD that is inspired by a variant of the classical singular spectrum analysis (SSA) approach for time series by combining it with the classical cumulative sum (CUSUM) statistic from sequential hypothesis testing. In particular, we model the underlying dynamics of multivariate time series observations through the spatio-temporal model introduced recently in the multivariate SSA (mSSA) literature. The change point in such a setting corresponds to a change in the underlying spatio-temporal model. As the primary contributions of this work, we develop an algorithm based on CUSUM-statistic to detect such change points in an online fashion. We extend the analysis of CUSUM statistics, traditionally done for the setting of independent observations, to the dependent setting of (multivariate) time series under the spatio-temporal model. Specifically, for a given parameter $h > 0$, our method achieves the following desirable trade-off: when a change happens, it detects it within $O(h)$ time delay on average, while in the absence of change, it does not declare false detection for at least $\exp(\Omega(h))$ time length on average. We conduct empirical experiments using benchmark and synthetic datasets. We find that the proposed method performs competitively or outperforms the state-of-the-art change point detection methods across datasets.

## 1 Introduction

The task of change point detection (CPD) is concerned with detecting significant changes in the temporal evolution of a system from noisy observations. This task has attracted considerable attention in the statistics and machine learning communities as it has a broad range of applications including quality control [5], climate research [35], and speech recognition [36]. This interest resulted in many methods and algorithms-we refer to [3] for a thorough review.

Despite the relatively simple setup of the CPD problem, different variations of the problem have been studied and analyzed. We contrast two general settings characterized by the assumptions on the data generating process. The first is the classical view that aims to detect distribution changes from independent and identically distributed (i.i.d.) observations, cf. [34] and more recently [7, 23]. The second deals with the more intricate time series dynamics by assuming the data to be dependent and following a specific temporal structure. Therein, the aim is to detect changes in this temporal structure of the observations. Some of the time series models considered include auto-regressive models [4, 46] and state-space models [20].

An important model comes from the singular spectrum analysis (SSA) literature, where the data is assumed to follow an underlying function associated with a low-dimensional subspace. This time series model has inspired algorithms for different inference tasks in time series, including imputation, forecasting, and CPD [13]. The key idea behind using SSA for CPD is that the subspace spanned by the columns of the

35th Conference on Neural Information Processing Systems (NeurIPS 2021).

Hankel matrix constructed from an initial subsequence of the time series is approximately equivalent to the one spanned by the later subsequences of the time series. The theoretical analysis of this algorithm focused primarily on the probability of error in the asymptotic setting where the signal and noise are assumed to be separable. However, previous work on SSA for CPD does not provide finite-sample analysis on any metric on interest in CPD. Further, its extension to the case of multivariate time series (multiple SSA or mSSA) lacks theoretical understanding, despite being empirically effective [11]. In this work, we consider a variant of mSSA proposed and analyzed in [1] for the tasks of imputation and forecasting. We extend the use of this mSSA variant to devise an algorithm for CPD.

Another important distinction between CPD methods lies in whether the method is *online* versus *offline*. In the offline setting, the dataset is fixed and the detection is made retrospectively based on the knowledge of the entire dataset. In such settings, only the accuracy of the detection is of concern. In contrast, in the online setting, the data arrives sequentially and a decision must be made as quickly as possible. In addition to the detection accuracy, the delay in making the detection (i.e. the number of data points observed after the change until it is detected) is another important metric in the online setting.

A common overarching framework for the online CPD problem extends from sequential hypothesis testing, where the problem is formulated as a binary hypothesis test that is applied sequentially as new data comes in [40]. This framework provides a way to evaluate the performance of online CPD methods in terms of the trade-off between false alarm rate and detection delay. The classical setting of this framework uses parametric models and forms likelihood ratio-based test statistics. In more recent work, this framework has been utilized with different test statistics. For example, tests using statistics based on the kernel Fisher discriminant ratio [16], maximum mean discrepancy [25], and subspace distance [19] were proposed. However, rigorously characterizing the trade-off between false alarm rate and detection delay for time series is absent in the literature.

In this work, we are motivated to fill the gap in literature by providing an *online* method for CPD for the setting of *multivariate time series data* (not i.i.d.) to detect change points with *provable* performance guarantees and *desirable empirical* performance.

**Summary of contributions.** Below, we summarize the main contributions of this work.

1. We introduce an expressive spatio-temporal model as described in Section 2.1 to model the dynamics and posit the question of detection as a change in these dynamics.
2. We develop a variant of the SSA algorithm for online change point detection in multivariate time series. In particular, we propose an algorithm that utilizes the low-dimensional structure of the time series to construct a cumulative sum (CUSUM) statistic (a la [34]) based on subspace distance. We refer to Section 3 for a full description of the proposed algorithm.
3. We analyze the performance of this algorithm in terms of the average running length (ARL), a common metric used in CUSUM tests to measure both the delay in detection and the average running time until a false alarm. Our main result shows that in the case of no change, the expected running length is exponential in the selected threshold (Theorem 4.1). While in the presence of change, the expected detection delay is linear in the threshold (Theorem 4.2). The results are consistent with the behavior of the CUSUM tests in the i.i.d. case.
4. We conduct comparative empirical experiments against state-of-the-art change point detection methods using benchmark datasets. In Table 1, we provide a summary that suggests that our method is at least as good, if not better than, state-of-art methods (SOTA) across datasets.

**Related work.** Due to space constraints, we provide an overview of related topics in Appendix A.

Table 1: The F1-score improvement our algorithm achieves compared to SOTA methods. See section 5 for details.

| Benchmark Data | Improvement over SOTA | Synthetic Data | Improvement over SOTA |
|---|---|---|---|
| Beedance | 10.4% | Energy | 8.2% |
| Hasc | 7.6% | Mean | 14.1% |
| Occupancy | 65.2% | Mixed | 51.4% |
| Yahoo | 0.6% | Frequency | 36.1% |

## 2  Problem Setup

We consider the discrete time setting where for each time index $t \in [T] := \{1, ... T\}$, we observe a multivariate time series $\mathbf{X}(t) := [X_1(t), ..., X_N(t)] \in \mathbb{R}^N$. For each $n \in [N]$, we denote the $n$-th latent time series by

$f_n : \mathbb{Z}^+ \to \mathbb{R}$, such that at each timestep $t \in [T]$, the observations take the form $X_n(t) = f_n(t) + e_n(t)$, where $e_n(t)$ is the per-step noise modeled as a mean-zero i.i.d. random variable. We define the latent multivariate time series as $\mathbf{f} : \mathbb{Z}^+ \to \mathbb{R}^N$ such that $\mathbf{f}(t) := [f_1(t), ..., f_N(t)]$. That is, the observation $\mathbf{X}(t)$ takes the form

$$\mathbf{X}(t) = \mathbf{f}(t) + \mathbf{e}(t), \tag{1}$$

where $\mathbf{e}(t) := [e_1(t), ..., e_n(t)]$. We note that although the additive noise $e_n(t)$ is i.i.d., the latent time series $\mathbf{f}(t)$ can have a complex dependence structure across both $t$ and $n$, see Section 2.1.

Our objective is to detect changes in the latent time series $\mathbf{f}(t)$ as quickly as possible. Precisely, our goal is to detect a change point $\tau$ such that

$$\mathbf{X}(t) = \begin{cases} \mathbf{f_0}(t) + \mathbf{e}(t) & \text{if } t < \tau \\ \mathbf{f_1}(t) + \mathbf{e}(t) & \text{if } t \geq \tau. \end{cases}$$

where $\mathbf{f_0}, \mathbf{f_1} : \mathbb{Z}^+ \to \mathbb{R}^N$ and $\mathbf{f_0}(t) \neq \mathbf{f_1}(t)$. In this setting, we treat the change point $\tau$ as an unknown deterministic quantity. We frame this CPD problem as a sequential hypothesis test where at each time $t$ one of two hypotheses is accepted

$$\begin{aligned} H_0 &: \mathbb{E}[\mathbf{X}(t)] = \mathbf{f_0}(t) \\ H_1 &: \mathbb{E}[\mathbf{X}(t)] = \mathbf{f_1}(t). \end{aligned} \tag{2}$$

Let $H(t) : \mathbb{Z}^+ \to \{0, 1\}$ be the indicator random variable that indicates whether or not the null hypothesis is rejected at time t. We then estimate the change point $\tau$ as

$$\hat{\tau} = \inf\{t | H(t) = 1\}. \tag{3}$$

Ideally, we wish to have $\hat{\tau} \geq \tau$, but also $\hat{\tau} - \tau$ to be small.

## 2.1 Time Series Model

There are multiple settings for the CPD problem characterized by the probabilistic structure of the observations. In this work, we consider the spatio-temporal factor model for multivariate time series introduced in [1], which captures a wide variety of time series dynamics. Next, we describe the model in detail. The spatio-temporal factor model requires the underlying latent multivariate time series to satisfy two properties. The first property captures the "spatial" structure, i.e., the structure across the $N$ time series (Property 2.1); and the second property captures the "temporal" structure (Property 2.2).

**Property 2.1** (Spatial structure). *There exist $R \in \mathbb{N}$, where $1 \leq R \ll \min(N, T)$, $W_r : \mathbb{Z}^+ \to \mathbb{R}$, and $\alpha_{nr} \in \mathbb{R} \; \forall r \in [R]$, $n \in [N]$, such that for any $n \in [N]$, $t \in [T]$, $f_n(t) = \sum_{r=1}^{R} \alpha_{nr} W_r(t)$, where $|\alpha_{nr}| \leq \Gamma_\alpha$, $|W_r(t)| \leq \Gamma_W$ for constants $\Gamma_\alpha$, $\Gamma_W > 0$.*

Property 2.1 implies that there are $R$ "fundamental" time series, and each latent time series $f_n(\cdot) \; \forall n \in [N]$ can be obtained through a weighted combination of these $R$ times series. To capture the temporal structure, additional assumptions are imposed on these fundamental time series $W_r(\cdot) \; \forall r \in [R]$. To do so, we first introduce the Page matrix representation of a time series.

**Definition 2.1** (Page Matrix). *Given a time series $f : \mathbb{Z}^+ \to \mathbb{R}$, its Page matrix representation over $T$ observations with parameter $1 \leq L \leq T$ is given by the matrix $\mathbf{Z} \in \mathbb{R}^{L \times \lfloor T/L \rfloor}$ with $Z_{ij} = f(i + (j - 1) \times L)$ for $i \in [L]$, $j \in [\lfloor T/L \rfloor]$.*

**Property 2.2** (Temporal structure). *For each $r \in [R]$ and for any $T > 1$, $1 \leq L \leq T$, let $\mathbf{Z}_W^{(r)}$ be $\mathbb{R}^{L \times \lfloor T/L \rfloor}$ Page Matrix associated with time series $W_r(t)$, $t \in [T]$. Then, the rank of $\mathbf{Z}_W^{(r)}$ is at most $G$ for any choice of $1 \leq L \leq T$.*

Property 2.2 imposes a particular temporal structure in the time series $W_r(\cdot), \forall r \in [R]$. While seemingly restrictive at first, it has been shown in the SSA literature that many standard functions that model time series dynamics satisfy this property. In particular, the Page matrix representation of any finite sum of products of harmonics, low-degree polynomials, and exponential functions is low-rank and satisfies Property 2.2 (refer to Proposition 2.1 in [1]). These functions can capture a rich class of time series dynamics which are typically modeled by three components: stationarity, periodicity, and trend. Periodicity is modeled as harmonics and trend as polynomials; the mixture of both are instances of this model. The inclusion of stationarity in this

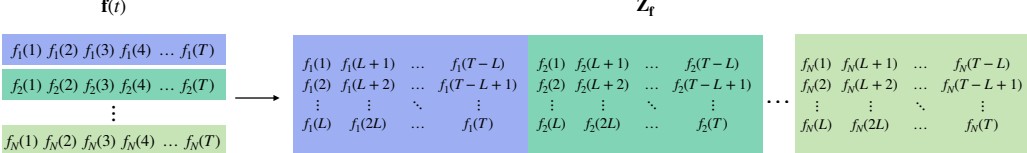

Figure 1: A depiction of the stacked Page matrix of the multivariate time series $\mathbf{f}(t)$.

model can be understood by considering the spectral representation of stationary processes (refer to Property 4.1 in [39]), which states that any sample-path of a stationary process can be decomposed into a sum of harmonics. Therefore, a finite sum of harmonics provides a good model representation for stationary processes.

For a given $1 \leq L \leq T$, let $\mathbf{Z}_f^{(n)} \in \mathbb{R}^{L \times \lfloor T/L \rfloor}$ denote the Page matrix induced by the $n$-th latent time series $f_n(t) \; \forall n \in [N]$. Further, consider the stacked Page matrix $\mathbf{Z_f} \in \mathbb{R}^{L \times (N \lfloor T/L \rfloor)}$, obtained by a column-wise concatenation of the Page matrices $\mathbf{Z}_f^{(1)},...,\mathbf{Z}_f^{(N)}$ (See Figure 1). Specifically,

$$\mathbf{Z_f} = \begin{bmatrix} \mathbf{Z}_f^{(1)} & \mathbf{Z}_f^{(2)} & ... & \mathbf{Z}_f^{(N)} \end{bmatrix}. \tag{4}$$

Proposition 2.1 (Proposition 2.2 in [1]) establishes that under the spatio-temporal factor model satisfying Properties 2.1 and 2.2, the stacked Page matrix is low-rank. We refer to [1] for the proof of this property.

**Proposition 2.1** (Proposition 2.2 in [1]). *Let Properties 2.1 and 2.2 hold. Then for any $1 \leq L \leq T$, the rank of $\mathbf{Z}_f^{(n)}$ for $n \in [N]$ is at most $R \times G$. Moreover, the rank of the stacked Page matrix $\mathbf{Z_f}$ is also at most $R \times G$.*

### 2.2 Changes in the Spatio-Temporal Factor Model

In this section, we give a characterization of the change points in view of the spatio-temporal model described in Section 2.1. Let us consider the following interpretation of Proposition 2.1: for a multivariate time series $\mathbf{f}(\cdot)$ that satisfies Properties 2.1 and 2.2, and for a sufficiently large $L$ and number of observations $T$, the columns of its stacked Page matrix span the same linear subspace of $\mathbb{R}^L$ independently of $T$. To formalize the notion of sufficiently large $L$ and $T$, we introduce the definition of the order of the time series.

**Definition 2.2** (Order of the time series). *Given a multivariate time series $\mathbf{f} : \mathbb{Z}^+ \to \mathbb{R}^N$, we define its order as the integer $k > 0$ such that for all $L \geq k$ and $T$ such that $N \times \lfloor T/L \rfloor \geq k$, the stacked Page matrix representation of the time series with parameter $L$ over $T$ observations $\mathbf{Z_f} \in \mathbb{R}^{L \times N \lfloor T/L \rfloor}$ has rank $k$.*

Proposition 2.1 suggests that a time series that follows Properties 2.1 and 2.2 has a finite order $k \leq R \times G$. Further, any $L, T$ that satisfy $L \geq k$, $N \times \lfloor T/L \rfloor \geq k$, are sufficient for this to hold. In lieu of the above, we can characterize changes in the latent time series using the distance from the subspace spanned by the columns of the stacked Page matrix $\mathbf{Z_f}$. Before we do so, we introduce appropriate notations and definitions.

**Definition 2.3** (L-lagged vectors). *Given a time series $f : \mathbb{Z}^+ \to \mathbb{R}$ and a lag parameter $L > 1$, the $L$-lagged vector at time $t \in \{L,...,T\}$ is given by $f(t-L+1:t) := [f(t-L+1),...,f(t)] \in \mathbb{R}^L$.*

For a multivariate time series $\mathbf{f}(t)$, we define the $L$-lagged matrix at time $t \in \{L,...,T\}$ as the column-wise concatenation of the $L$-lagged vectors $f_n(t-L+1:t)$ for $n \in [N]$. Specifically,

$$\mathbf{f}(t-L+1:t) := [f_1(t-L+1:t) \quad ... \quad f_N(t-L+1:t)]. \tag{5}$$

We analogously denote the $L$-lagged observation matrix at time $t$ by $\mathbf{X}(t-L+1:t)$. Let $\lambda_1(\mathbf{f}(t-L+1:t))$ be the largest singular values of the $L$-lagged matrix of $\mathbf{f}(\cdot)$ at time $t \in \{L,...,T\}$. We define the following quantities for the pre-change functions $\mathbf{f_0}(\cdot)$:

$$\lambda_1^{\max,0} := \max_{t < \tau} \lambda_1^2(\mathbf{f_0}(t-L+1:t)), \qquad \lambda_1^{\min,0} := \min_{t < \tau} \lambda_1^2(\mathbf{f_0}(t-L+1:t), \tag{6}$$

with $\lambda_1^{\max,1}$ and $\lambda_1^{\min,1}$ analogously defined for $\mathbf{f_1}(\cdot)$ over the time range $t \geq \tau + L - 1$. Now let $\mathbb{L}_0 := \mathsf{span}(\mathbf{Z_{f_0}}) \subset \mathbb{R}^L$, where $\mathbf{Z_{f_0}}$ is the stacked Page matrix induced by $\mathbf{f_0}(\cdot)$ with parameter $L$ over $T_0$ observations (assuming no change occurs at any $t \leq T_0$). If the latent time series continues to follow the same function for $t > T_0$, then the columns of its $L$-lagged matrix belong to the subspace $\mathbb{L}_0$[1]. The

---

[1]Since Proposition 2.1 is stated for the Page matrix representation, this is true for $L$-lagged vectors that are non-overlapping and spaced by $L$ observations. However, an analogous result to Proposition 2.1 can be proven for the Hankel matrix representation making this valid for any segment of $L$ observations. For simplicity, we omit discussion of the Hankel matrix but we refer to [1] for such results.

occurrence of the change point at $t=\tau$ forces some or all of the columns of the the $L$-lagged matrix to leave the pre-change subspace $\mathbb{L}_0$. Eventually, i.e., for $t \geq \tau + L - 1$, the columns of the $L$-lagged matrix will be in the post change subspace $\mathbb{L}_1$ which corresponds to the function $\mathbf{f_1}(\cdot)$. Following this characterization of change points, we revisit our hypotheses in (2) and reformulate them as

$$H_0 : \mathsf{span}(\mathbb{E}[\mathbf{X}(t-L+1:t)]) \subset \mathbb{L}_0 \tag{7}$$
$$H_1 : \mathsf{span}(\mathbb{E}[\mathbf{X}(t-L+1:t)]) \not\subset \mathbb{L}_0.$$

The occurrence of a change point can be captured by a test statistic that monitors the distance between the lagged vectors and the pre-change subspace. This motivates the CPD algorithm we propose next.

## 3 Algorithm

The core steps in the proposed CPD algorithm are to first estimate the pre-change subspace from the stacked Page matrix of noisy observations $\mathbf{X}(t)$, then repeatedly estimate the distance between this subspace and future lagged vectors. If our estimation of the subspace is accurate, the distance will remain small as long as the observations continue to follow the same latent time series. Once a change occurs, the lagged vectors are forced to leave the subspace causing the distance to become larger. The algorithm uses the subspace distance as a detection score to construct a CUSUM statistic to perform the hypothesis test in (7). In what follows, we review the CUSUM procedure for sequential hypothesis testing then we introduce the full CPD algorithm.

### 3.1 CUSUM Procedure

**Definition 3.1** (CUSUM Statistic). *Given an observation $\mathbf{X}(t)$ we assign a detection score $D(t)$, and define the CUSUM statistic of the observations $\{\mathbf{X}(1),...,\mathbf{X}(t)\}$ as $y(t) := \max_{1 \leq i \leq t} \sum_{j=i}^{t} D(j)$*

It is easy to verify that the CUSUM statistic in Definition 3.1 can be, more conveniently, computed recursively (also known as Lindley's recursion in queueing theory [26]) as

$$y(t) = \max\{y(t-1) + D(t), 0\}, \qquad y(0) = 0. \tag{8}$$

Using the CUSUM statistic, and for a threshold $h > 0$, we define the decision rule at each time $t$ as

$$H(t) = H_{\mathbb{1}_{y(t) \geq h}},$$

and the estimated change point in (3) can be written as $\hat{\tau} = \inf\{t | y(t) \geq h\}$. The CUSUM procedure for change point detection, as introduced by Page [34] requires the assigned detection score $D(t)$ to satisfy the following Property 3.1.

**Property 3.1.** *A valid CUSUM detection score $D(t)$ has a negative expectation when no change is present, and a positive expectation when a change occurs, that is $\mathbb{E}[D(t) | H_0] < 0$ and $\mathbb{E}[D(t) | H_1] > 0$.*

### 3.2 Algorithm Description

Our mSSA algorithm for CPD has five parameters: (1) the number of initial observations $T_0$ used to estimate the subspace $\mathbb{L}_0$, (2) the lag parameter $1 < L \leq T_0$, (3) the estimated order of the time series $\hat{k} > 0$, (4) the CUSUM test threshold $h > 0$, and (5) the shift-downwards constant $c \geq 0$. For simplicity and without loss of generality[2], let $T_0$ be an integer multiple of $L$. We introduce two additional integer quantities: $M := T_0/L$ and $\bar{M} := N \times M$. In what follows, we detail the four main steps of the algorithm.

**Step I: Base Matrix Construction.** Transform the observations $X_n(1:T_0) := [X_n(1),...,X_n(T_0)], n \in [N]$ into a Page matrix $\mathbf{Z}_X^{(n)} \in \mathbb{R}^{L \times M}$ as per Definition 2.1. Then, form the stacked Page matrix $\mathbf{Z}_\mathbf{X} \in \mathbb{R}^{L \times \bar{M}}$ as in (4). We refer to $\mathbf{Z}_\mathbf{X}$ as the base matrix.

**Step II: Subspace Estimation.** Let $\mathbf{u}_i$ for $i \in [L]$ be the left singular vectors of the base matrix $\mathbf{Z}_\mathbf{X}$ sorted as per associated singular values in the decreasing order. We group the singular vectors into two matrices $\hat{\mathbf{U}}_0 = [\mathbf{u}_1,...,\mathbf{u}_{\hat{k}}]$ and $\hat{\mathbf{U}}_\perp = [\mathbf{u}_{\hat{k}+1},...,\mathbf{u}_L]$. Then, we estimate the pre-change subspace $\mathbb{L}_0$ as:

$$\hat{\mathbb{L}}_0 = \mathsf{span}(\hat{\mathbf{U}}_0). \tag{9}$$

Let $\hat{\mathbb{L}}_\perp = \mathsf{span}(\hat{\mathbf{U}}_\perp)$ be the orthogonal complement of the subspace $\hat{\mathbb{L}}_0$ in $\mathbb{R}^L$.

---

[2]If $T_0$ is not integer multiple of $L$ we can introduce a new $T_0' = L \times \lfloor T_0/L \rfloor$.

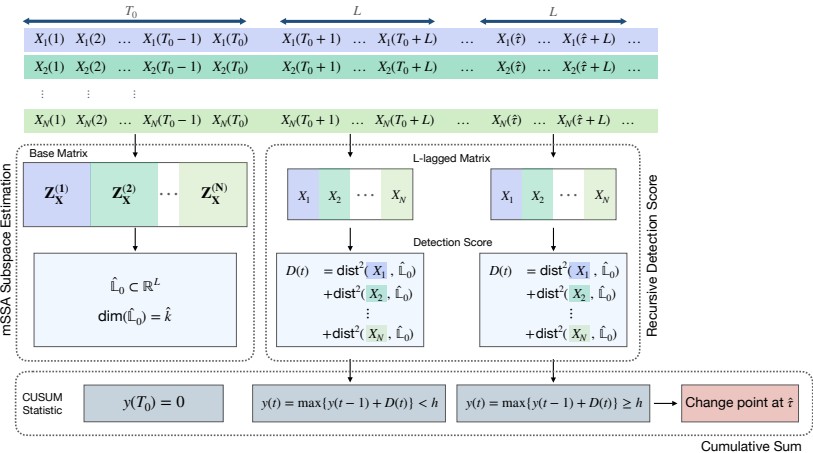

Figure 2: Visual illustration of the algorithm steps.

**Step III: Detection Scores.** For $t > T_0$, construct the L-lagged matrix $\mathbf{X}(t-L+1:t)$ analogous to (5). We define the subspace detection score, which measure the Euclidean distance between the columns of the L-lagged matrix and the estimated subspace $\hat{\mathbb{L}}_0$, as

$$D(t) = \|\hat{\mathbf{U}}_\perp^T \mathbf{X}(t-L+1:t)\|_F^2 - c. \tag{10}$$

**Step IV: CUSUM Test.** For $t > T_0$, compute the CUSUM statistic $y(t)$ using (8) with $y(T_0) = 0$.

Steps III and IV are repeated until a change point is detected when $y(t) > h$. That is,

$$\hat{\tau} = \inf_{t > T_0} \{t \,|\, y(t) \geq h\} \tag{11}$$

To adapt this algorithm for multiple change point detection, when a change point is detected at $\hat{\tau}$, the algorithm restarts with a new base matrix being constructed using the segment $\mathbf{X}(\hat{\tau}:\hat{\tau}+T_0-1)$. Refer to Figure 2 for a visual depiction of these steps, and Algorithm 1 in Appendix C for a summary of the algorithm.

### 3.3 Choice of Parameters

Below, we provide general guidelines for selecting the parameters $T_0$ and $L$, $\hat{k}$. Further, we detail a data-driven procedure for choosing the CUSUM parameters $c$ and $h$ in Appendix D.1.

A general recommendation is to choose $T_0$ to be as large as possible and set $L = \lfloor \sqrt{\min(N, T_0) \times T_0} \rfloor$ to achieve a good estimation of the pre-change subspace (see Proposition 4.1). However, if $T_0$ is too large we might potentially miss or smooth out a change point. In practice, one heuristic used to choose $T_0$ is as follows: given a starting $T_0$, construct the stacked Page matrix and compute its "effective rank" (e.g. top $k$ singular values whose sum of squares is at least 90% of the squared Frobenius norm). Then keep increasing $T_0$ till the "effective rank" of $\mathbf{Z_X}$ stops changing. Once it does, this indicates that we have captured the principal subspace $\mathbb{L}_0$. Additionally, the "effective rank" of matrix $\mathbf{Z_X}$ can be used as $\hat{k}$, the estimated order of time series. Indeed, it is well understood that the largest singular value of a random $A \times B$ matrix with independent zero mean sub-Gaussian entries scales as $O(\sqrt{A} + \sqrt{B})$. That is, the $L \times \bar{M}$ matrix $\mathbf{Z_X}$, which can be viewed as a summation of the 'signal' matrix (corresponding to the stacked time series $\mathbf{f_0}(\cdot)$) and 'noise' matrix (corresponding to the stacked observation noise $\mathbf{e}(\cdot)$), is such that the top $k$ singular values scale as $\Theta(\sqrt{L \times \bar{M}})$ while other singular values scale as $O(\sqrt{L} + \sqrt{\bar{M}})$. Therefore, with large enough $L$ and $\bar{M}$, a clear separation is observed and is the reason for this heuristic to work. See [8] and [10] for use of such reasoning and heuristic in the context of matrix estimation literature.

## 4 Theoretical Analysis

In this section, we establish theoretical performance guarantees of the described algorithm. To that end, we formally introduce the performance objective. Recall that if a change point is present at time $t = \tau$, then

$$\mathbf{f}(t) = \begin{cases} \mathbf{f_0}(t) & \text{if } t < \tau \\ \mathbf{f_1}(t) & \text{if } t \geq \tau, \end{cases}$$

where $\mathbf{f_0}(t)$ and $\mathbf{f_1}(t)$ each satisfying Properties 2.1 and 2.2 with parameters $R_i, G_i, \Gamma_{\alpha i}$, and $\Gamma_{Wi}$ for $i \in \{0, 1\}$. The algorithm declares $\hat{\tau}$ as the estimate of $\tau$. Therefore, ideally if $\tau < \infty$ then we wish to have $\hat{\tau} \geq \tau$, and $\hat{\tau} - \tau$ as small as possible. Further, if $\tau = \infty$, i.e. no change point is present, then we want $\hat{\tau}$ to be as large as possible. This is captured through *average running length* of type 0 and type 1 defined next.

**Average running length.** This metric is typically considered in the setting of sequential hypothesis testing [34]. Let $T_1 = T_0 + 1$, i.e., one time-step after the time horizon used for the base matrix. Then, define

$$ARL_0 = \mathbb{E}[\hat{\tau} - T_0 | \tau = \infty] \equiv \mathbb{E}_\infty[\hat{\tau}], \quad ARL_1 = \mathbb{E}[\hat{\tau} - T_0 | \tau = T_1] \equiv \mathbb{E}_{T_1}[\hat{\tau}].$$

The desired behavior is to have a large $ARL_0$ and small $ARL_1$. In what follows, we provide bounds on these two quantities for the specific choice of the detection score $D(t)$ as a function of the threshold $h$.

**Subspaces distance and affinity.** We introduce the definitions of two quantities that will be used in the main results. The first is $\epsilon$ which describes how well is our estimation of the pre-change subspace. The other is $\delta$, which describes the similarity between the pre and post change subspaces. Recall the subsapces $\mathbb{L}_0, \mathbb{L}_1, \hat{\mathbb{L}}_0$, and $\hat{\mathbb{L}}_\perp$, and their corresponding basis matrices. Then define

$$\epsilon \equiv \mathsf{dist}(\mathbb{L}_0, \hat{\mathbb{L}}_0) := \|\hat{\mathbf{U}}_\perp^T \mathbf{U}_0\|_{op}, \quad \delta \equiv \mathsf{affinity}(\mathbb{L}_0, \mathbb{L}_1) := \|\mathbf{U}_0^T \mathbf{U}_1\|_{op} \tag{12}$$

**Operating assumptions.** We make certain operating assumptions to establish our results stated next.

**Assumption 4.1** (Parameter Selection). *We assume that the parameters of the algorithm are chosen such that $T_0 \leq \tau$, $L = \lfloor\sqrt{\min(N, T_0) \times T_0}\rfloor$, $L \leq \bar{M}$, and $L \geq k$.*

**Assumption 4.2** (Gaussian Noise). *We shall assume i.i.d. Gaussian noise $e_n(t) \sim N(0, \sigma^2)$.*[3]

**Assumption 4.3** (Balanced spectra). *Let $\mathbf{Z_f}$ be the stacked Page matrix associated with the time series $\mathbf{f}(\cdot)$ that satisfies properties 2.1 and 2.2 and has order $k$, where under the setup of Proposition 2.1, $k \leq R \times G$. Then for $L = \lfloor\sqrt{\min(N, T_0) \times T_0}\rfloor \geq k$, the stacked page matrix is such that $\lambda_k(\mathbf{Z_f}) \geq K\sqrt{NT_0}/\sqrt{k}$ for some absolute constant $K > 0$, where $\lambda_k(\cdot)$ is the $k$-th largest singular value of its argument.*

Based on the these assumptions, we can quantify the bound on the estimation error $\epsilon$, as follows.

**Proposition 4.1.** *Let assumptions 4.1-4.3 hold. Let $\epsilon$ be the estimation error defined in (12). Let*

$$\mathcal{E} := \{\epsilon \leq q\}, \quad \text{with} \quad q := \frac{4\sigma\sqrt{R_0 \times G_0}}{K(\min(T_0, N) \times T_0)^{1/4}}. \tag{13}$$

*Then $P(\mathcal{E}) \geq 1 - 2\exp(-2\sqrt{NT_0})$.*

Note that the choice of $L = \lfloor\sqrt{\min(N, T_0) \times T_0}\rfloor$ plays a key role in minimizing the subspace estimation error. Precisely, the bound on the estimation error ($q$) scales inversely with the square root of the minimum dimension of the Page matrix. Thus, to minimize the error, we construct a nearly square base matrix.

**Main results.** Now we state the main result providing a lower bound on $ARL_0$ and an upper bound on $ARL_1$ for a given parameter $h > 0$. In particular, it shows that $ARL_0$ scales exponentially in $h$ while $ARL_1$ scales linearly in $h$. Proofs of Proposition 4.1, Theorems 4.1, 4.2, and 4.3 can be found in Appendix B.

**Theorem 4.1** (Bound on $ARL_0$). *Let Assumptions 4.1-4.3 hold. Let $\hat{\tau}$ be the estimate of $\tau$ as per (11). Given $h > 0$, $\hat{k} = k$, and a choice of $c$ such that (recall notation in (6))*

$$c > N(L - k)\sigma^2 + qR_0(\lambda_1^{\max, 0})^2. \tag{14}$$

*Then, if $\tau = \infty$,*

$$ARL_0 = \mathbb{E}_\infty[\hat{\tau}] \geq C_0 \exp(C_1 h),$$

*where $C_0, C_1$ are positive constants that depend on the parameters $\sigma, R_0, G_0, N, L$ and $T_0$.*

**Theorem 4.2** (Bound on $ARL_1$). *Let Assumptions 4.1-4.3 hold. Let $\hat{\tau}$ be the estimate of $\tau$ as per (11). Conditioned on $\mathcal{E}$ and given $h > 0$, $\hat{k} = k$, and a choice of $c$ such that (recall notation in (6))*

$$c < N(L - k)\sigma^2 + (1 - \delta - q)(\lambda_1^{\min, 1})^2, \tag{15}$$

---

[3]The analysis and guarantees obtained will extend easily for generic sub-Gaussian distribution.

*if $\tau = T_0 + 1$, then*

$$ARL_1 \leq \tilde{C}_0 + \tilde{C}_1 h + \tilde{C}_2 \exp\left(-\tilde{C}_3 h\right), \tag{16}$$

*where $\tilde{C}_0$, $\tilde{C}_1$, $\tilde{C}_2$, and $\tilde{C}_3$ are positive constants that depend on the parameters $\sigma, c, N, L, R_1, \Gamma_{\alpha 1}, \Gamma_{W1}$.*

Next, we show that there exists a feasible choice of $c$ for both theorems to hold, provided that the pre and post change subspaces are sufficiently "different".

**Theorem 4.3.** *Let $\delta$ be the affinity between the pre and post change subspaces defined as per (12). Then there exists a feasible choice of $c$ that satisfies both equations (14) and (15) if $\delta$ is such that*

$$\delta < 1 - q\left(1 + R_0\left(\frac{\lambda_1^{\max,0}}{\lambda_1^{\min,1}}\right)^2\right). \tag{17}$$

*Interpretation.* Recall that $\delta$ is a measure of "similarity" between subspaces which takes a value in $[0,1]$: it is 1 if the two subspaces intersect, otherwise, it is $< 1$. Theorem 4.3 implies that as $q$ gets smaller, which may be achieved as more data points ($NT_0$) become available, we can tolerate larger $\delta$ without compromising the feasibility of $c$. Under perfect estimation (e.g. no noise in data or infinite data availability), i.e. q = 0, any $\delta < 1$ will guarantee the feasibility of $c$.

# 5 Evaluation on Benchmark Datasets

We present a comparative evaluation of two variants of the proposed mSSA algorithm and four other standard methods for CPD: BinSeg [21], KL-CPD [7], Microsoft SSA [29], and BOCPDMS [23].

**Datasets.** The evaluation is conducted on four benchmark datasets (see Table 2). Details of these datasets, their sources, and physical interpretations of change points are given in Appendix D.2.

**Evaluation metric.** For evaluation we use the F1-score metric, which is defined as: $F_1 = \frac{\text{TP}}{\text{TP} + \frac{1}{2}(\text{FP} + \text{FN})}$, where TP, FP, and FN denote the number of true positives, false positives, and false negatives, respectively. Following standard practice cf. [42, 27, 46], we use the following soft true positive (TP) rule: for a constant $\eta > 0$, if the algorithm detects a change point $\hat{\tau}$ such that $|\hat{\tau} - \tau| \leq \eta$ where $\tau$ is a labeled change point, a TP is recorded, otherwise a FP is recorded. Here we report the results for $\eta = 10$. To avoid double-counting, we ensure that each labeled and detected change point is associated with at most one TP. Further, we use the convention of counting the first index of the time series (t = 1) as a TP.

Note that we choose the F1-score as our metric over the average running length (i.e. $ARL_0$ and $ARL_1$) for two practical reasons: (i) it is a more commonly used metric in the change point detection literature; (ii) it can be used for both online and offline change point detection methods, both are used in our experiments.

**Practical implementation (mSSA-MW).** The Algorithm described in Section 3 is meant for a single change point detection. In practice, there can be multiple change points in a time series and hence we use a natural Moving Window variant. This is described in details in Algorithm 2 in Appendix C. In this section, we run the evaluations for both the original variant (mSSA) and the Moving Window variant (mSSA-MW).

Table 2: Summary of datasets.

|  | Dataset | Domain | No. time series | Length of time series | No. change points |
|---|---|---|---|---|---|
| Real-world data | Beedance | $\mathbb{R}^3$ | 6 | 608-1124 | 117 |
|  | HASC | $\mathbb{R}^3$ | 18 | 11738-12000 | 196 |
|  | Occupancy | $\mathbb{R}^4$ | 1 | 8143 | 12 |
|  | Yahoo | $\mathbb{R}$ | 99 | 1680 | 207 |
| Synthetic data | Energy | $\mathbb{R}$ | 20 | 5000 | 80 |
|  | Mean | $\mathbb{R}$ | 20 | 5000 | 80 |
|  | Frequency | $\mathbb{R}$ | 20 | 5000 | 80 |
|  | Mixed | $\mathbb{R}^{100}$ | 5 | 1000 | 20 |

**Parameter Selection.** When applying a CPD algorithm as a statistical tool for the automatic detection of change points, a common challenge that arises in practice is tuning the algorithm's parameters. This

is particularly challenging in the task of CPD as it is often applied in an unsupervised manner due to the scarcity of labels (change points are anomalous events). To overcome this, we follow the evaluation setup proposed in [42], where for each dataset we report (1) the F1-score using default parameters, and (2) the best F1-score over a grid search of parameter configurations. We provide details about the default parameters and the grid search for all algorithms in Appendix D.1.

**Results.** We compare with retrospective change point detection methods (BinSeg and KL-CPD) and real-time methods (BOCPDMS and Microsoft SSA). Two of these algorithms (BinSeg and Microsoft SSA) only support univariate time series. For datasets of higher dimension, we report the results of these algorithms on a univariate time series constructed from the L2 norm of the observation vector (in Appendix D.5, we explore other approaches of applying univariate algorithms to multivariate data).

In Table 3, the best and default performance of each algorithm are shown as the average and standard deviation of the F1-scores across time series in each dataset[4]. mSSA-MW algorithm shows consistently superior performance over other methods, including the retrospective ones, across all datasets in the best parameters setting, and all but one dataset using the default parameters.

For all datasets, we notice a significant improvement of mSSA-MW over the mSSA variant. We describe one possible explanation for this behavior. If the order of the time series is large compared to the dimensions of the base matrix, then the estimation of the subspace is inaccurate, leading to a large detection score even when no change occurs. While this is also true for the moving window approach, therein the base matrix estimates a new subspace in every time step, which captures the local structure of the time series at that time. While it doesn't necessarily capture the full signal, its estimated subspace has more proximity to the test lagged vectors. Another plausible explanation is the existence of gradual and small changes in the data that aren't labeled as change points. In such a case, constructing a fixed base matrix makes the algorithm more sensitive to such changes, while having a moving window smooths out such changes.

Table 3: Mean (and standard deviations) of F1-scores for each algorithm are reported. mSSA outperforms other methods across all datasets with best parameters choice. The No Change rows correspond to an algorithm that detects no change point.

| | Real-world Datasets | | | | | | | |
|---|---|---|---|---|---|---|---|---|
| | Beedance | | HASC | | Occupancy | | Yahoo | |
| | Best | Default | Best | Default | Best | Default | Best | Default |
| BinSeg | 0.597 (0.10) | 0.097 (0.02) | 0.304 (0.08) | 0.161 (0.05) | 0.308 (N/A) | 0.308 (N/A) | 0.715 (0.19) | 0.372 (0.10) |
| Microsoft SSA | 0.583 (0.06) | 0.279 (0.08) | 0.265 (0.07) | 0.049 (0.02) | 0.462 (N/A) | 0.375 (N/A) | 0.684 (0.19) | 0.283 (0.11) |
| KL-CPD | 0.401 (0.05) | 0.252 (0.07) | 0.156 (0.01) | 0.155 (0.01) | 0.341 (N/A) | 0.302 (N/A) | 0.699 (0.18) | **0.617 (0.20)** |
| BOCPDMS | 0.167 (0.07) | 0.092 (0.02) | 0.204 (0.06) | 0.078 (0.04) | 0.474 (N/A) | 0.198 (N/A) | 0.479 (0.19) | 0.463 (0.19) |
| **mSSA (ours)** | 0.404 (0.17) | 0.124 (0.04) | 0.190 (0.06) | 0.142 (0.03) | 0.526 (N/A) | 0.222 (N/A) | 0.667 (0.21) | 0.422 (0.18) |
| **mSSA MW (ours)** | **0.659** (0.12) | **0.500** (0.13) | **0.327** (0.10) | **0.177** (0.11) | **0.783** (N/A) | **0.480** (N/A) | **0.719** (0.18) | 0.384 (0.16) |
| No Change | 0.097 (0.02) | | 0.155 (0.01) | | 0.143 (N/A) | | 0.556 (0.22) | |
| | Synthetic Datasets | | | | | | | |
| | Energy | | Mean | | Mixed | | Frequency | |
| | Best | Default | Best | Default | Best | Default | Best | Default |
| BinSeg | 0.663 (0.18) | 0.209 (0.03) | 0.714 (0.18) | 0.600 (0.21) | 0.634 (0.14) | 0.240 (0.08) | 0.350 (0.03) | 0.200 (0.00) |
| Microsoft SSA | 0.849 (0.07) | **0.811** (0.09) | 0.807 (0.10) | **0.733** (0.17) | 0.546 (0.07) | 0.232 (0.03) | 0.735 (0.11) | 0.399 (0.26) |
| KL-CPD | 0.475 (0.12) | 0.399 (0.10) | 0.499 (0.12) | 0.407 (0.11) | 0.333 (0.00) | 0.333 (0.00) | 0.345 (0.05) | 0.333 (0.00) |
| BOCPDMS | 0.449 (0.13) | 0.280 (0.10) | 0.328 (0.09) | 0.213 (0.04) | 0.152 (0.02) | 0.131 (0.00) | 0.443 (0.07) | 0.209 (0.10) |
| mSSA (ours) | **0.929** (0.09) | 0.786 (0.21) | 0.915 (0.06) | 0.673 (0.28) | 0.950 (0.10) | 0.740 (0.17) | 0.994 (0.02) | 0.874 (0.19) |
| mSSA-MW (ours) | 0.919 (0.09) | 0.763 (0.17) | **0.921** (0.07) | 0.697 (0.31) | **0.960** (0.05) | **0.856** (0.10) | **1.000** (0.00) | **0.930** (0.11) |
| No Change | 0.333 (0.00) | | 0.333 (0.00) | | 0.333 (0.00) | | 0.333 (0.00) | |

## 6  Evaluation on Synthetic Data

To further explore its performance, we evaluate mSSA on synthetic datasets, using the same setup of Section 5. We follow the practice of [7] and [27] and create synthetic datasets using our proposed time series model, each with a different representative mode of change. Specifically, we generate four sets

---

[4]For the occupancy, we only have one time series, and hence the standard deviation is not reported.

of synthetic time series, each is a mixture of harmonics and a polynomial trend with the following modes of change: (i) jumping mean (Mean), a univariate time series where at each change point, a constant offset is added; (ii) scaling signal energy (Energy), a univariate time series where at each change point, the amplitude of the harmonics is changed; (iii) changing frequency (Frequency), a univariate time series where at each change point, the frequencies of the harmonics are changed; (iv) mixed change (Mixed), a multivariate time series where at each change point, a mixture of the above changes is applied. Refer to Table 2 for more details and Appendix D.3 for details about the data generation process.

**Results.** In Table 3, the best and default performance of each algorithm are shown as the average and standard deviation of the F1-scores across time series in each synthetic dataset. Both mSSA and mSSA-MW implementations show consistently superior performance over other methods across all datasets in the best parameters setting, and they compete with Microsoft SSA in the default parameters setting. Notice that the results on the simulated data show a similar performance of the two variants of the algorithm (mSSA and mSSA-MW). Here the data is constructed to follow the proposed spatio-temporal model with a low-rank Page matrix, and enough time steps are available to construct a sufficiently large base matrix. As a result, the subspaces of the fixed base matrix (in mSSA) and the moving window base matrices (in mSSA-MW) become almost identical with small variations due to the noise.

For the univariate datasets, another algorithm that performs well is the Microsoft SSA, which is proposed for a similar time series model. However, we notice a significant degradation in the performance in the case of the multivariate time series. This highlights the importance of extending SSA to accommodate multivariate time series. The BinSeg algorithm performs relatively well on the Energy and Mean datasets. However, the algorithm is not capable of detecting the Frequency mode of change, nor the Mixed mode. We note that BinSeg has designated methods for detecting mean and variance changes, which might be the reason for its good performance in the Energy and Mean datasets. KL-CPD and BPCPDMS algorithms are built with the assumption of i.i.d. samples, so the highly dependant structure and the time-varying mean of our data might explain the bad performance of these algorithms on the simulated datasets.

**Multivariate data helps.** Here, we explore the benefits of utilizing information *across* time series in mSSA. Specifically, we study the trade-off between $ARL_0$ and $ARL_1$ as $N$ (number of time series) increases. We start by generating 25 synthetic time series as described in details in Appendix D.4. Then for each value of $N \in [5, 10, 15, 20, 25]$ we perform the following experiment. First, we assume that there is no change point ($\tau = \infty$) and run the algorithm for 10 trials. In each trial, we vary the threshold $h$ and record $\hat{\tau}$ which represents the time of false alarm. Next, we introduce a change point at $T_0 + 1$, and run the algorithm for another 10 trials. Again we vary the threshold $h$ and record $\hat{\tau}$ which represents the detection delay.

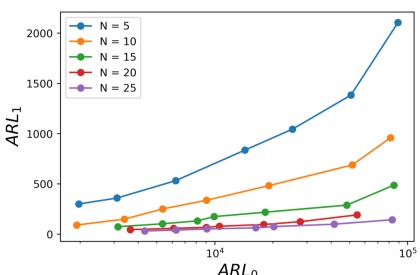

Figure 3: Trade-off between $ARL_0$ and $ARL_1$ as we increase the dimension of the time series.

Figure 3 shows the average results of the 10 trials for each pair $(N, h)$. Each line of connected dots corresponds to the same value of $N$ with different values of $h$. As shown in the figure, as we fix the value of $ARL_0$ at a certain acceptable rate, we can achieve better $ARL_1$ (i.e. detection delay) using a time series of a larger dimension. This shows that mSSA is effectively utilizing information across multiple time series to improve the detection performance over the univariate SSA.

## 7 Limitations

Our method achieves encouraging theoretical and empirical results for change point detection. However, like any such research program, there are various limitations suggesting exciting directions for future research. A prominent one being extending the theoretical analysis to the setting of *multiple* change points. In a sense, this would be important step forward for the field of change point or anomaly detection.

## Acknowledgements and Funding Disclosure

This work was supported in parts by the MIT-IBM project on time series anomaly detection, NSF TRIPODS Phase II grant towards Foundations of Data Science Institute, KACST project on Towards Foundations of Reinforcement Learning, and scholarship from KACST (for Arwa Alanqary and Abdullah Alomar).

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
