# A  Related Work

The literature on CPD is vast. Here, we provide an overview of two very closely related topics to the current work. First, CPD methods that are based on singular spectrum analysis and its extension for multivariate time series. Second, the cumulative sum (CUSUM) approach for change point detection and sequential hypothesis testing.

**Singular Spectrum Analysis.** Singular spectrum analysis (SSA) is a subspace-based method for time series analysis that has a well developed theory and solves a wide array of problems, including CPD. It overcomes many limitations of time series analysis, such as nonlinearity and nonstationarity of the signal. The main steps of the original SSA method are: (1) embedding the time series into a Hankel matrix, (2) singular value decomposition of the Hankel matrix, (3) grouping the singular triplets to separate the different components of the time series. The key idea behind using SSA for CPD is that the subspace spanned by the columns of the Hankel matrix constructed from a subsequence of the time series is approximately equivalent to the one spanned by the later subsequences of the time series. A comprehensive introduction to the theoretical basis of the SSA method for CPD can be found in [13]. An algorithm for CPD based on the original SSA was proposed by Moskvina and Zhigljavsky [31]. The theoretical analysis of this algorithm focused on the probability of error in the asymptotic setting where the signal and noise are assumed to be separable. A modification to this algorithm was proposed by Mohammad and Nishida [30] to increase robustness under very noisy conditions. Following this work, several applications of SSA for CPD has emerged [18, 41].

Multivariate singular spectrum analysis (mSSA) is an extension of the original SSA for simultaneous analysis of multiple time series that share a common structure [17]. The steps of mSSA are the same as those of the original SSA except that the first step is done by stacking the Hankel matrices of the individual time series. Despite the empirical success of mSSA in the tasks of forecasting and imputation, variants of this algorithm for CPD are very limited and lack theoretical analysis [11]. A variant of the mSSA algorithm was introduced by Agarwal et al. [1] for performing the tasks of forecasting and imputation. It utilizes the Page matrix representation instead of the Hankel matrix in step (1). This variant is a lot simpler compared to the original mSSA, has stronger theoretical guarantees, and as good or better empirical performance for time series imputation and forecasting [2, 1]. In the present work, we develop an algorithm based on this variant of mSSA to characterize and track changes in multivariate time series.

**CUSUM Tests & Sequential Hypothesis Testing.** Extending ideas from sequential hypothesis testing to address the problem of CPD emerged in the 1940s. The classical setup was that of quickest detection of an unknown point of change from one parametric probability distribution to another in a stream of i.i.d. data. The aim is to detect the change-point with minimal delay under constraints on false detections. This classical setting of the problem has been considered by Shewhart [38], Girshick & Rubin [12] and Page [34].

The cumulative sum (CUSUM) test proposed by Page [34] is particularly ubiquitous in the literature of CPD, due to its simplicity and strong theoretical guarantees. Performance of the CUSUM test is often evaluated using the average running length (ARL), which captures a trade-off between the false alarm rate and the delay in detecting a change point. While optimality results of the CUSUM approach for the classical setting of CPD were established by Moustakides [32] in the 1980s, further variants of the CUSUM test have been proposed and analyzed for various settings of the CPD problem. For example, several recent works, including that of Lee et al. [24], propose CUSUM-based methods to tackle change points in observations generated from well-known time series models, including ARMA-GARCH models. While the work of Lee et al. [24] proves the consistency of the proposed estimator, it does not provide finite-sample analysis of the ARL, and it makes restrictive parametric assumptions about the data generating process. Another recent study by Jiao et al. [19] proposes a CUSUM-based algorithm for detecting changes while assuming a low-dimensional latent structure of a stream of high-dimensional data. This work derives finite-sample performance bounds in terms of the ARL. However, these results are based on the assumption that the observations are i.i.d.. In the present work, we utilize a spatio-temporal model for multivariate time series data, and we use the mSSA algorithm within the CUSUM framework for addressing the CPD problem. We analyze the performance of our proposed algorithm in terms of the ARL under the assumed spatio-temporal model.

# B  Proofs of Results in Section 4

## B.1  Definitions and Helper Lemmas

**Multivariate Gaussian random vectors.** We introduce the definition and some known results on multivariate Gaussian vectors.

**Definition B.1** (Multivariate Gaussian vectors [14]). *A random vector $\mathbf{x} \in \mathbb{R}^n$ has a multivariate Gaussian distribution if it can be expressed in the form*

$$\mathbf{x} = \mathbf{D}\mathbf{z} + \mu$$

*for some matrix $\mathbf{D} \in \mathbb{R}^{n \times n}$ and some real vector $\mu \in \mathbb{R}^n$ where $\mathbf{z} \in \mathbb{R}^n$ is a random vector whose components are independent standard normal random variables.*

**Lemma B.1** ([14]). *The components of a multivariate Gaussian vector are uncorrelated if and only if they are independent.*

**Lemma B.2** ([28]). *If $\mathbf{A} \in \mathbb{R}^{n \times n}$ is a real symmetric matrix, $\mathbf{X} \in \mathbb{R}^n$ is a multivariate Gaussian with mean vecotr $\mu \in \mathbb{R}^n$ and covariance matrix $\mathbf{\Sigma} \in \mathbb{R}^{n \times n} > 0$ and $\mathbf{B} \in \mathbb{R}^n$ is a constant vector, then for $Q = \mathbf{X}^{\mathbf{T}}\mathbf{A}\mathbf{X} + \mathbf{B}^{\mathbf{T}}\mathbf{X}$ and any $x \in \mathbb{R}$*

$$g_Q(x) = \mathbb{E}[\exp(x\,Q)] = |\mathbf{I} - 2x\mathbf{A}\mathbf{\Sigma}|^{-\frac{1}{2}} \exp\left(-\frac{1}{2}\left(\mu^T \mathbf{\Sigma}^{-1} \mu\right) + \frac{1}{2}(\mu + x\mathbf{\Sigma}\boldsymbol{B})^T (\mathbf{I} - 2x\mathbf{A}\mathbf{\Sigma})^{-1} \mathbf{\Sigma}^{-1}(\mu + x\mathbf{\Sigma}\boldsymbol{B})\right).$$

**Moment generating functions (MGFs).** We introduce a property of the logarithm of the MGF of a random variable.

**Lemma B.3** ([14]). *Let $g_X(x) = \mathbb{E}[\exp(xX)]$ be the moment generating function of a random variable $X \in \mathbb{R}$ defined on a set $D \subset \mathbb{R}$ such that $0 \in D$. Then $\ln(g_X(x))$ is convex in $x \in D$ and hence $g_X(x)$ is continuous in the interior of $D$.*

**Sub-Gaussian and sub-exponential random variables.** We introduce the definitions and some classic properties of sub-Gaussian and sub-exponential random variables.

**Definition B.2** (Sub-Gaussian random variable [44]). *A random variable $X \in \mathbb{R}$ is said to be sub-Gaussian with variance proxy $\sigma^2$ (denoted as $X \sim subG(\sigma^2)$) if $\mathbb{E}[X] = 0$ and its moment generating function satisfies*

$$g_X(x) = \mathbb{E}[\exp(xX)] \leq \exp\left(\frac{\sigma^2 x^2}{2}\right) \forall x \in \mathbb{R}$$

**Definition B.3** (Sub-exponential random variable [44]). *A random variable $Y \in \mathbb{R}$ is said to be sub-exponential with parameter $\nu$ (denoted as $Y \sim subE(\nu)$) if $\mathbb{E}[X] = 0$ and its moment generating function satisfies*

$$g_X(x) = \mathbb{E}[\exp(xX)] \leq \exp\left(\frac{\nu^2 x^2}{2}\right) \forall |x| \leq \frac{1}{\nu}$$

**Lemma B.4** ([43, 44]). *Let $X \sim subG(\sigma_s^2)$ and $Z \sim N(0, \sigma_g^2)$. For $i \in [N]$, let $Y_i \sim subE(\nu_i)$. Then:*

1. *$Z \sim subG(\sigma_g^2)$*

2. *$Z \sim subE(\sigma_g)$*

3. *$X^2 - \mathbb{E}[X^2] \sim subE(16\sigma_s^2)$*

4. *$\sum_{i=1}^N Y_i \sim subE(\sum_{i=1}^N \nu_i)$*

*If $Y_i$ are independent:*

5. *$\sum_{i=1}^N Y_i \sim subE((\sum_{i=1}^N \nu_i^2)^{1/2})$*

**Concentration inequalities.** We introduce some known concentration inequalities for scalar random variables and random matrices.

**Lemma B.5** ([43]). *Let $\mathbf{A} \in \mathbb{R}^{n \times n}$ be a real matrix whose entries are independent standard normal random variables. Then for every $b \geq 0$, with probability at least $1 - 2\exp(-b^2/2)$*

$$\lambda_{\max}(\mathbf{A}) \leq \sqrt{n} + \sqrt{m} + b,$$

where $\lambda_{\max}(\mathbf{A})$ is the largest singular value of $\mathbf{A}$.

**Lemma B.6** ([22]). *Let $X_1, X_2, ..., X_N \in \mathbb{R}$ be independent non-negative random variables with $\mathbb{E}[X_i] \leq 1 \; \forall i \in [N]$, then for any $\epsilon > 0$*

$$P\left(\prod_{i=1}^{N} X_i \geq \epsilon\right) \leq \frac{1}{\epsilon}$$

**Lemma B.7** (Bernstein inequality [44]). *Let $Y_i \sim subE(\nu_i)$. If $Y_i$ are independent and $\nu_1 = \cdots = \nu_N = \nu$ then*

$$P\left(\sum_{i}^{N} Y_i > \epsilon\right) \leq exp\left(-\frac{1}{2}\left(\frac{\epsilon^2}{N\nu^2} \wedge \frac{\epsilon}{\nu}\right)\right)$$

## B.2 Proof of Proposition 4.1

By direct application of Wedin's $\sin(\theta)$ Theorem (see [9, 45]), we can bound $\epsilon$ as follows. Recall that $\mathbf{Z_f}$ is the stacked Page matrix of $\mathbf{f_0}(1:T_0)$ and that $\text{rank}(\mathbf{Z_f}) = k$, then

$$\epsilon \leq \frac{\|\mathbf{E}\|_{op}}{\lambda_k(\mathbf{Z_f})}, \tag{18}$$

where $\mathbf{E}$ is the stacked Page matrix of the additive noise $\mathbf{e}(1:T_0)$. For simplicity and without loss of generality[5], let us assume that $\sqrt{\min(N,T_0) \times T_0} \in \mathbb{Z}^+$. By helper Lemma B.5 we have

$$\begin{aligned}
\|\mathbf{E}\|_{op} &\leq \sigma(\sqrt{\bar{M}} + \sqrt{L} + 2(N \times T_0)^{1/4}) \\
&\leq \sigma(2\sqrt{\bar{M}} + 2(N \times T_0)^{1/4}) \\
&= 2\sigma\left[(\max(N,T_0) \times N)^{1/4} + (N \times T_0)^{1/4}\right],
\end{aligned} \tag{19}$$

with probability at least $1 - 2\exp(-2\sqrt{NT_0})$, where the second inequality follows from the assumption $L = \sqrt{\min(N,T_0) \times T_0} \leq \bar{M}$, and the last equality follows from

$$\begin{aligned}
\bar{M} &= N\frac{T_0}{L} \\
&= N\frac{T_0}{\sqrt{\min(N,T_0) \times T_0}} \\
&= \begin{cases} \sqrt{N \times T_0} & \text{if } N < T_0 \\ N & \text{if } N \geq T_0 \end{cases} \\
&= \sqrt{\max(N,T_0) \times N}.
\end{aligned}$$

Next, and as a direct consequence of Assumption 4.3, we have

$$\lambda_k(\mathbf{Z_f}) \geq \frac{K\sqrt{N \times T_0}}{\sqrt{R_0 \times G_0}}. \tag{20}$$

By plugging (19) and (20) in (18) we get

$$\begin{aligned}
\epsilon &\leq \frac{2\sigma\sqrt{R_0 \times G_0}(\max(N,T_0) \times N)^{1/4}}{K\sqrt{N \times T_0}} + \frac{2\sigma\sqrt{R_0 \times G_0}(N \times T_0)^{1/4}}{K\sqrt{N \times T_0}} \\
&= \frac{2\sigma\sqrt{R_0 \times G_0}}{K}\left(\frac{1}{\sqrt{\min(T_0, \sqrt{N \times T_0})}} + \frac{1}{(N \times T_0)^{1/4}}\right) \\
&= \frac{2\sigma\sqrt{R_0 \times G_0}}{K\sqrt{\min(T_0, \sqrt{N \times T_0})}}\left(1 + \sqrt{\frac{\min(T_0, \sqrt{N \times T_0})}{\sqrt{N \times T_0}}}\right) \\
&= \begin{cases} \frac{4\sigma\sqrt{R_0 \times G_0}}{K(N \times T_0)^{\frac{1}{4}}} & \text{if } N < T_0 \\ \frac{2\sigma\sqrt{R_0 \times G_0}}{K\sqrt{T_0}}\left(1 + \sqrt{\frac{T_0}{N}}\right) & \text{if } N \geq T_0 \end{cases} \\
&\leq \frac{4\sigma\sqrt{R_0 \times G_0}}{K(\min(T_0,N) \times T_0)^{1/4}} =: q,
\end{aligned} \tag{21}$$

with probability at least $1 - 2\exp(-2\sqrt{NT_0})$.

---

[5]We can always adjust the choice of $T_0$ to make this true.

## B.3 Proof of Theorem 4.1

Recall from Proposition 4.1 the definition of the event $\mathcal{E}$ and that $P(\mathcal{E}) \geq 1 - 2\exp(-2\sqrt{NT_0})$. Then, by definition of the $ARL_0$, we can write

$$
\begin{aligned}
ARL_0 &= \mathbb{E}[\hat{\tau} - T_0 \,|\, \tau = \infty] \\
&= \mathbb{E}[\hat{\tau} - T_0 \,|\, \tau = \infty, \mathcal{E}]P(\mathcal{E}) + \mathbb{E}[\hat{\tau} - T_0 \,|\, \tau = \infty, \mathcal{E}^c]P(\mathcal{E}^c) \\
&\geq \mathbb{E}[\hat{\tau} - T_0 \,|\, \tau = \infty, \mathcal{E}]P(\mathcal{E}) \\
&\geq \mathbb{E}[\hat{\tau} - T_0 \,|\, \tau = \infty, \mathcal{E}]\left(1 - 2\exp\left(-2\sqrt{NT_0}\right)\right).
\end{aligned}
\tag{22}
$$

For the rest of the proof we will find a lower bound on $\mathbb{E}[\hat{\tau} - T_0 \,|\, \tau = \infty, \mathcal{E}]$ using the following steps.

**Step I. Moment generation function of the detection score.** As a first step, we derive an analytical expression of the moment generating function (MGF) of $D(t)$ at each time step $t > T_0$.

First, recall that $\mathbf{u}_i$ for $i \in [L]$ denote the left singular vectors of the base matrix $\mathbf{Z_X}$ sorted as per associated singular values in the decreasing order. Also recall that $\hat{\mathbf{U}}_\perp = [\mathbf{u}_{k+1}, ..., \mathbf{u}_L] \in \mathbb{R}^{L \times (L-k)}$. Recall that the detection score is defined as:

$$
\begin{aligned}
D(t) &= \|\hat{\mathbf{U}}_\perp^T \mathbf{X}(t-L+1:t)\|_F^2 - c \\
&= \|\hat{\mathbf{U}}_\perp^T \mathbf{f}(t-L+1:t) + \hat{\mathbf{U}}_\perp^T \mathbf{e}(t-L+1:t))\|_F^2 - c,
\end{aligned}
$$

where the second equality follows from (1). For ease of exposition, we introduce the notation $\mathbf{B}(t) := \hat{\mathbf{U}}_\perp^T \mathbf{e}(t-L+1:t)$ and

$$
\mathbf{A}(t) := \begin{cases} \hat{\mathbf{U}}_\perp^T \mathbf{f_0}(t-L+1:t) & \text{for } t < \tau, \\ \hat{\mathbf{U}}_\perp^T \mathbf{f_1}(t-L+1:t) & \text{for } t \geq \tau+L-1. \end{cases}
\tag{23}
$$

Further, let $\mathbf{A}_n(t) \in \mathbb{R}^{(L-k)}$ and $\mathbf{B}_n(t) \in \mathbb{R}^{(L-k)}$ for $n \in [N]$ denote the $n$-th column of $\mathbf{A}(t) \in \mathbb{R}^{(L-k) \times N}$ and $\mathbf{B}(t) \in \mathbb{R}^{(L-k) \times N}$, respectively. With that, we can rewrite the detection score as follows:

$$
\begin{aligned}
D(t) &= \|\mathbf{A}(t) + \mathbf{B}(t)\|_F^2 - c \\
&= \|\mathbf{A}(t)\|_F^2 + \|\mathbf{B}(t)\|_F^2 + 2\,\text{Trace}\left(\mathbf{A}(t)^T \mathbf{B}(t)\right) - c \\
&= \sum_{n \in [N]} \mathbf{A}_n(t)^T \mathbf{A}_n(t) + \sum_{n \in [N]} \mathbf{B}_n^T(t) \mathbf{B}_n(t) + 2 \sum_{n \in [N]} \mathbf{A}_n^T(t) \mathbf{B}_n(t) - c.
\end{aligned}
$$

To characterize the distribution of the detection score, we start with the random vector $\mathbf{B}_n(t)$. Note that the components of this vector are $\mathbf{u}_i^T e_n(t-L+1:t)$ for $i = k+1, ..., L$ and thus are the weighted sum of independent Gaussian random variables. Hence, each component is distributed as $N(0, \|\mathbf{u}_i\|_2^2 \sigma^2)$. But recall that $\|\mathbf{u}_i\|_2 = 1 \,\forall i \in [L]$, and hence, the components are distributed as $N(0, \sigma^2)$.

Further for each $i \neq j$, $\text{cov}\left(\mathbf{u}_i^T e_n(t-L+1:t), \mathbf{u}_j^T e_n(t-L+1:t)\right) = 0$ since components of $e_n(t-L+1:t)$ are independent, $\mathbb{E}[e_n(\cdot)] = 0$, and $\mathbf{u}_i^T \mathbf{u}_j = 0$. According to definition B.1, the vector $\mathbf{B}_n(t)$ is a multivariate Gaussian with mean vector $\mu = \mathbf{0}$ and covariance matrix $\Sigma = \text{diag}(\sigma^2)$. Note that $\mathbf{A}_n$ is a deterministic quantity for all $n \in [N]$, as it is based on a fixed realization of the base matrix.

Now, consider the random variable $D_n(t) = \mathbf{B}_n^T(t)\mathbf{B}_n(t) + 2\mathbf{A}_n^T(t)\mathbf{B}_n(t)$. By helper Lemma B.2, the MGF of $D_n(t)$ is

$$
g_{D_n(t)}(x) = \left(1 - 2x\sigma^2\right)^{-\frac{L-k}{2}} \exp\left(\frac{2x^2\sigma^2}{1 - 2x\sigma^2}\|\mathbf{A}_n(t)\|_2^2\right)
$$

Going back to the detection score, we can write it in terms of $D_n(t)$ as:

$$
D(t) = \sum_{n \in [N]} \|\mathbf{A}_n(t)\|_2^2 + \sum_{n \in [N]} D_n(t) - c.
$$

By utilizing the independence of $D_n(t)$ across $n$, which follows from the independence of the additive noise $e_n(\cdot)$ across $n$, we derive the MGF of $D(t)$ as:

$$
g_{D(t)}(x) = \exp\left(\sum_{n \in [N]} \|\mathbf{A}_n(t)\|_2^2 x - cx\right) \prod_{n \in [N]} g_{D_n(t)}(x)
$$

$$= (1 - 2x\sigma^2)^{-\frac{N(L-k)}{2}} \exp\left(\frac{x}{1 - 2\sigma^2 x} \|\mathbf{A}(t)\|_F^2 - cx\right), \tag{24}$$

which is defined for $x \in (-\infty, 1/2\sigma^2)$.

**Step II. Expectation of the detection score.** We can directly obtain the expectation of $D(t)$ as the first derivative of the MGF in (24) evaluated at $x = 0$, which gives

$$\mathbb{E}[D(t)] = N(L-k)\sigma^2 + \|\mathbf{A}(t)\|_F^2 - c. \tag{25}$$

**Step III. Detection score has negative expectation.** Here we show that for any choice of $c$ as indicated in the theorem statement, $\mathbb{E}[D(t)|\mathcal{E}] < 0 \ \forall t > T_0$.

To do so, let us begin by finding an upper bound on $\mathbb{E}[D(t)|\mathcal{E}] \ \forall t > T_0$ by bounding $\max_{t > T_0} \|\mathbf{A}(t)\|_F$. Recall that for $t < \tau$, $f_{0,n}(t - L + 1 : t) \in \mathbb{L}_0 \ \forall n$. Let $\mathbf{U}_0$ be a matrix whose columns are orthonormal basis of $\mathbb{L}_0$. Then for some matrix $\mathbf{S}(t) \in \mathbb{R}^{k \times N}$ we can write the lagged vectors and the expression of $\mathbf{A}(t)$ in (23) as:

$$\mathbf{A}(t) = \hat{\mathbf{U}}_\perp^T \mathbf{f_0}(t - L + 1 : t)$$
$$= \hat{\mathbf{U}}_\perp^T \mathbf{U}_0 \mathbf{S}(t),$$

for all $T_0 < t < \tau$. Notice that $\|\mathbf{S}(t)\|_F = \|\mathbf{f_0}(t - L + 1 : t)\|_F$. Now we give an upper bound on $\max_{t > T_0} \|\mathbf{A}(t)\|_F^2$

$$\max_{t > T_0} \|\hat{\mathbf{U}}_\perp^T \mathbf{U}_0 \mathbf{S}(t)\|_F^2 \leq \|\hat{\mathbf{U}}_\perp^T \mathbf{U}_0\|_{op}^2 \max_{t > T_0} \|\mathbf{S}(t)\|_F^2$$
$$= \|\hat{\mathbf{U}}_\perp^T \mathbf{U}_0\|_{op}^2 \max_{t > T_0} \|\mathbf{f_0}(t - L + 1 : t)\|_F^2$$
$$\leq \|\hat{\mathbf{U}}_\perp^T \mathbf{U}_0\|_{op} R_0 \max_{t > T_0} \lambda_1^2(\mathbf{f_0}(t - L + 1 : t))$$
$$= \|\hat{\mathbf{U}}_\perp^T \mathbf{U}_0\|_{op} R_0 \lambda_1^{\max,0},$$

where the third line follows by the fact that $\|\hat{\mathbf{U}}_\perp^T \mathbf{U}_0\|_{op} \leq 1$ and that each column in $\mathbf{f_0}(t - L + 1 : t)$ is a linear combination of $R_0$ fundamental time series (Property 2.1), thus $\text{rank}(\mathbf{f_0}(t - L + 1 : t)) \leq R_0$. In the last line we use the notation from (6). Notice that $\|\hat{\mathbf{U}}_\perp^T \mathbf{U}_0\|_{op} = \epsilon$, so conditioned on $\mathcal{E}$ we have

$$\max_{t > T_0} \|\hat{\mathbf{U}}_\perp^T \mathbf{U}_0 \mathbf{S}(t)\|_F^2 \leq q R_0 \lambda_1^{\max,0}. \tag{26}$$

Thus, by plugging (26) in the (25) we get

$$\mathbb{E}[D(t)|\mathcal{E}] \leq N(L-k)\sigma^2 + q R_0 \lambda_1^{\max,0} - c$$

So any choice of $c$ as described in the theorem statement will make $\mathbb{E}[D(t)|\mathcal{E}] < 0 \ \forall t > T_0$.

**Step IV. Bound on the tail probability of the CUSUM statistic.** Define the notation of the probability $P_\infty(\cdot) := P(\cdot | \tau = \infty, \mathcal{E})$. Recall from Definition 3.1 and for any $t > T_0$

$$y(t) = \max_{T_0 < j \leq t} \left(\sum_{i=j}^t D(i)\right).$$

Let $t^* = \text{argmax}_{T_0 < j \leq t}\left(\sum_{i=j}^t D(i)\right)$, then

$$P_\infty(y(t) \geq h) = P_\infty\left(\sum_{i=t^*}^t D(i) \geq h\right)$$
$$= P_\infty\left(\sum_{\ell=1}^L \sum_{j=0}^{I(\ell)-1} D(t^* - 1 + j \times L + \ell) \geq h\right)$$

In the last equality, we grouped the detection scores into $L$ *independent* groups, where $I(\ell)$ for $\ell \in [L]$ denotes the number of observations in the $\ell$-th group. Specifically, $I(\ell)$ is defined as follows:

$$I(\ell) = \begin{cases} \lfloor (t-t^*)/L \rfloor + 1 & \text{if } \ell \leq (t-t^*+1) \pmod{L} \\ \lfloor (t-t^*)/L \rfloor & \text{if } \ell > (t-t^*+1) \pmod{L}. \end{cases}$$

Note that the event $\sum_{i=t^*}^t D(i) > h$ implies at least one of the events $\sum_{j=0}^{I(\ell)-1} D(t^*-1+j \times L+\ell) \geq h/L$ for some $\ell \in [L]$. That is,

$$P_\infty(y(t) \geq h) \leq P_\infty \left( \bigcup_{\ell=1}^L \left( \sum_{j=0}^{I(\ell)-1} D(t^*-1+j \times L+\ell) \geq h/L \right) \right)$$

$$\leq \sum_{\ell=1}^L P_\infty \left( \sum_{j=0}^{I(\ell)-1} D(t^*-1+j \times L+\ell) \geq h/L \right)$$

$$= \sum_{\ell=1}^L P_\infty \left( \prod_{j=0}^{I(\ell)-1} \exp(xD(t^*-1+j \times L+\ell)) \geq \exp(xh/L) \right)$$

for any $x > 0$. If $x$ is selected such that $\mathbb{E}[\exp(xD(i))] = g_{D(t)}(x) \leq 1 \; \forall i \in [t^*, t]$, then by helper lemma B.6 we can show that

$$P_\infty(y(t) \geq h) \leq L \exp\left(-\frac{xh}{L}\right). \tag{27}$$

**Step V. Find $x^* > 0$ such that $g_{D(t)}(x^*) \leq 1$ for all $t$.** Define the logarithm of the MGF

$$M_{D(t)}(x) = \ln(g_{D(t)}(x)). \tag{28}$$

Then the condition $g_{D(t)}(x) \leq 1$ is equivalent to $M_{D(t)}(x) \leq 0$. Let us highlight the following two properties of $M_{D(t)}(\cdot)$:

- It is convex in the interval $x \in (-\infty, 1/2\sigma^2)$ by helper Lemma B.3.
- Its derivative satisfies $\frac{d}{dx}M_{D(t)}(0) = \frac{d}{dx}g_{D(t)}(0)$, which can be easily verified.

Note that when conditioned on $\mathcal{E}$, $\frac{d}{dx}g_{D(t)}(0) = \mathbb{E}[D(t) \mid \mathcal{E}] < 0$ for all $t > T_0$. Thus, we can find $x^* > 0$ such that $M_{D(t)}(x^*) \leq 0$ for all $t > T_0$ by solving the following optimization problem:

$$\begin{aligned} \min \quad & \max_{t > T_0} M_{D(t)}(x) \\ \text{s.t.} \quad & x > 0 \\ & x < \frac{1}{2\sigma^2}. \end{aligned} \tag{29}$$

We solve this problem by finding a minima of the function $\max_t M_{D(t)}(x)$ then showing that it is in the interval $(0, 1/2\sigma^2)$. We begin by solving for the roots of

$$\frac{d}{dx}\max_t M_{D(t)}(x) = \frac{N(L-k)\sigma^2(1-2\sigma^2 x) + \max_{t > T_0}\|\mathbf{A}(t)\|_F^2 - c(1-2\sigma^2 x)^2}{(1-2\sigma^2 x)^2}.$$

Note that the numerator is a polynomial in $x$ and has a root at

$$x' = \frac{2c - N(L-k)\sigma^2 - \sqrt{N^2(L-k)^2\sigma^4 + 4c\max_{t > T_0}\|\mathbf{A}(t)\|_F^2}}{4c\sigma^2}.$$

We now show that this $x'$ satisfies the constraints in (29). First, we show that $x' > 0$, which requires

$$2c - N(L-k)\sigma^2 > \sqrt{N^2(L-k)^2\sigma^4 + 4c\max_{t > T_0}\|\mathbf{A}(t)\|_F^2},$$

or equivalently, by squaring both sides of the inequality,

$$N(L-k)\sigma^2 + \max_{t>T_0}\|\mathbf{A}(t)\|_F^2 - c < 0$$

Recall that for our choice of $c$, $\mathbb{E}[D(t)|\mathcal{E}] = N(L-k)\sigma^2 + \max_t\|\mathbf{A}(t)\|_F^2 - c < 0$. The second constraint requires

$$x' = \frac{1}{2\sigma^2} - \frac{N(L-k)}{4c} - \frac{\sqrt{N^2(L-k)^2\sigma^4 + 4c\max_{t>T_0}\|\mathbf{A}(t)\|_F^2}}{4c\sigma^2} < \frac{1}{2\sigma^2},$$

which follows by the fact that the second and third terms on the left hand side are strictly positive. Now let us replace $\max_{t>T_0}\|\mathbf{A}(t)\|_F^2$ with its bound derived in (26) and define

$$x^* = \frac{2c - N(L-k)\sigma^2 - \sqrt{N^2(L-k)^2\sigma^4 + 4cqR_0\max_t\lambda_1^2(\mathbf{f}_0(t-L+1:t))}}{4c\sigma^2}, \tag{30}$$

such that $x^* \leq x'$. So it is sufficient to show that $x^* > 0$ for it to be a valid choice that satisfies $g_{D(t)}(x^*) \leq 1$. For this we require

$$2c - N(L-k)\sigma^2 > \sqrt{N^2(L-k)^2\sigma^4 + 4cqR_0\max_t\lambda_1^2(\mathbf{f}_0(t-L+1:t))},$$

or equivalently, by squaring both sides of the inequality,

$$c > N(L-k)\sigma^2 + qR_0\max_t\lambda_1^2(\mathbf{f}_0(t-L+1:t)),$$

which is satisfied by the condition on the choice of $c$ in the theorem statement.

**Step VI. Bound on $\mathbb{E}[\hat{\tau} - T_0 | \tau = \infty, \mathcal{E}]$.** For any $T'$, we have

$$\mathbb{E}[\hat{\tau} - T_0 | \tau = \infty, \mathcal{E}] = \sum_{t'=1}^{\infty} t' P_\infty(\hat{\tau} - T_0 = t')$$

$$\geq T' \sum_{t'=\lceil T'\rceil+1}^{\infty} P_\infty(\hat{\tau} - T_0 = t')$$

$$= T'\left(1 - \sum_{t'=1}^{\lceil T'\rceil} P_\infty(\hat{\tau} - T_0 = t')\right). \tag{31}$$

Further, we can bound $\sum_{t'=1}^{\lceil T'\rceil} P_\infty(\hat{\tau} - T_0 = t')$ as follows:

$$\sum_{t'=1}^{\lceil T'\rceil} P_\infty(\hat{\tau} - T_0 = t') \leq \sum_{t'=1}^{\lceil T'\rceil} P_\infty(y(T_0+t') \geq h)$$

$$\leq \sum_{t'=1}^{\lceil T'\rceil} L\exp\left(-\frac{x^*h}{L}\right)$$

$$= \lceil T'\rceil L\exp\left(-\frac{x^*h}{L}\right)$$

$$\leq (T'+1)L\exp\left(-\frac{x^*h}{L}\right), \tag{32}$$

where the first inequality follows from the definition of $\hat{\tau}$ in (11) and the second follows from (27). By plugging (32) in (31) we get

$$\mathbb{E}[\hat{\tau} - T_0 | \tau = \infty, \mathcal{E}] \geq T'\left(1 - (T'+1)L\exp\left(-\frac{x^*h}{L}\right)\right). \tag{33}$$

Recall that this is valid for any $T'$. By choosing $T' = \frac{1}{2L}\exp(\frac{x^*h}{L}) - 1$ in (33) we get

$$\mathbb{E}[\hat{\tau} - T_0 | \tau = \infty, \mathcal{E}] \geq \frac{1}{4L}\exp\left(\frac{x^*h}{L}\right) - \frac{1}{2} \tag{34}$$

Finally, by plugging (34) back in (22) we get

$$ARL_0 \geq \frac{1}{4L} \exp\left(\frac{x^* h}{L}\right) \left(1 - 2\exp\left(-2\sqrt{NT_0}\right)\right) - \frac{1}{2}$$

Note that $ARL_0 \geq 1$, and by appropriate choice of constants $C_0, C_1$, the $-\frac{1}{2}$ term can be subsumed in the following lower bound,

$$ARL_0 \geq C_0 \exp(C_1 h),$$

which completes the proof.

## B.4 Proof of Theorem 4.2

**Step I. Distribution of the centered detection score.** As a first step in the proof we show that at each $t > T_0 + L$, the detection score $D(t)$, after centering, has a sub-exponential distribution and derive its parameter.

First, recall that $\mathbf{u}_i$ for $i \in [L]$ denote the left singular vectors of the base matrix $\mathbf{Z_X}$ sorted as per associated singular values in the decreasing order. Also recall that $\hat{\mathbf{U}}_\perp = [\mathbf{u}_{k+1}, ..., \mathbf{u}_L] \in \mathbb{R}^{L \times (L-k)}$. Recall that the detection score is defined as:

$$D(t) = \|\hat{\mathbf{U}}_\perp^T \mathbf{X}(t-L+1:t)\|_F^2 - c$$
$$= \|\hat{\mathbf{U}}_\perp^T \mathbf{f}(t-L+1:t) + \hat{\mathbf{U}}_\perp^T \mathbf{e}(t-L+1:t))\|_F^2 - c,$$

where the second equality follows from (1). As was done in Appendix B.3, we use the notation $\mathbf{B}(t)$ and $\mathbf{A}(t)$ to simplify the exposition. Recall that $\mathbf{B}(t) := \hat{\mathbf{U}}_\perp^T \mathbf{e}(t-L+1:t)$ and

$$\mathbf{A}(t) := \begin{cases} \hat{\mathbf{U}}_\perp^T \mathbf{f}_0(t-L+1:t) & \text{for } t < \tau, \\ \hat{\mathbf{U}}_\perp^T \mathbf{f}_1(t-L+1:t) & \text{for } t \geq \tau + L - 1. \end{cases} \tag{35}$$

And recall that $\mathbf{A}_n(t) \in \mathbb{R}^{(L-k)}$ and $\mathbf{B}_n(t) \in \mathbb{R}^{(L-k)}$ for $n \in [N]$ denote the $n$-th column of $\mathbf{A}(t) \in \mathbb{R}^{(L-k) \times N}$ and $\mathbf{B}(t) \in \mathbb{R}^{(L-k) \times N}$, respectively. With that, we can rewrite the detection score as follows:

$$D(t) = \|\mathbf{A}(t) + \mathbf{B}(t)\|_F^2 - c$$
$$= \|\mathbf{A}(t)\|_F^2 + \|\mathbf{B}(t)\|_F^2 + 2 \operatorname{Trace}(\mathbf{A}(t)^T \mathbf{B}(t)) - c.$$

Recall the expectation of the detection score derived in (25). Now we consider the centered random variable

$$D'(t) = D(t) - \mathbb{E}[D(t)] \tag{36}$$
$$= (\|\mathbf{A}(t)\|_F^2 + \|\mathbf{B}(t)\|_F^2 + 2 \operatorname{Trace}(\mathbf{A}(t)^T \mathbf{B}(t)) - c) - (N(L-k)\sigma^2 + \|\mathbf{A}(t)\|_F^2 - c)$$
$$= \|\mathbf{B}(t)\|_F^2 + 2 \operatorname{Trace}(\mathbf{A}(t)^T \mathbf{B}(t)) - N(L-k)\sigma^2$$
$$= \sum_{n \in [N]} (\|\mathbf{B}_n(t)\|_2^2 + 2\mathbf{A_n}(t)^T \mathbf{B_n}(t) - (L-k)\sigma^2) \equiv \sum_{n \in [N]} D'_n(t),$$

such that $D'_n(t), \forall n \in [N]$ is defined as

$$D'_n(t) = \|\mathbf{B}_n(t)\|_2^2 + 2\mathbf{A_n}(t)^T \mathbf{B_n}(t) - (L-k)\sigma^2.$$

We aim to show that $D'(t)$ follows a sub-exponential distribution and derive its parameter. To do so, we first show that each $D'_n(t)$ is a sub-exponential random variable by the following steps:

- Let $B_n^i(t)$ for $i \in [L-k]$ denote the $i$-th component of $\mathbf{B}_n(t)$. Note that $B_n^i(t) = \mathbf{u}_{i+k}^T e_n(t-L+1:t)$ and recall from Step I in Appendix B.3 that $B_n^i(t) \sim N(0, \sigma^2)$.
- Recall also from Step I in Appendix B.3 that the components of $\mathbf{B}_n(t)$ are uncorrelated, thus, by helper Lemma B.1, they are independent.
- Then, by parts (1) and (3) of Lemma B.4, we have that $B_n^i(t)^2 - \sigma^2 \sim \mathrm{subE}(16\sigma^2)$
- Then, by part (5) of Lemma B.4, we have that the sum of these components is also sub-exponential. That is, $\|\mathbf{B}_n(t)\|_2^2 - (L-k)\sigma^2 \sim \mathrm{subE}(16\sigma^2 \sqrt{L-k})$.
- The random variable $2\mathbf{A}_n^T(t)\mathbf{B}_n(t) \sim N(0, 4\|\mathbf{A}_n(t)\|_2^2 \sigma^2)$.
- Then, by part (2) of Lemma B.4, we have $2\mathbf{A}_n^T(t)\mathbf{B}_n(t) \sim \mathrm{subE}(2\|\mathbf{A}_n(t)\|_2 \sigma)$.
- By part (4) of Lemma B.4 we have $D'_n(t) \sim \mathrm{subE}(16\sigma^2 \sqrt{L-k} + 2\|\mathbf{A}_n(t)\|\sigma)$.

Finally, we can apply part (4) of Lemma B.4 again to show that

$$D'(t) = \sum_{n \in [N]} D'_n(t) \sim \mathrm{subE}(16\sigma^2 N \sqrt{L-k} + 2\sigma \sum_{n \in [N]} \|\mathbf{A}_n(t)\|_2).$$

Herein, we use $\nu_t := 16\sigma^2 N\sqrt{L-k}+2\sigma\sum_{n\in[N]}\|\mathbf{A}_n(t)\|_2$, to denote the sub-exponential parameter of $D'(t)$.

**Step II. Expectation of the detection score is bounded below.** Here we want to show that for any choice of $c$ as indicated in the theorem statement, $\mathbb{E}[D(t)\,|\,\mathcal{E}]>0\ \forall t\geq T_0+L$.

To do so, let us begin by finding a lower bound on $\mathbb{E}[D(t)\ |\ \mathcal{E}]\ \forall t\geq T_0+L$ by bounding $\min_{t\geq T_0+L}\|\mathbf{A}(t)\|_F$. Recall that for $t\geq T_0+L$, $f_{1,n}(t-L+1:t)\in\mathbb{L}_1\ \forall n$. Let $\mathbf{U}_1$ be a matrix whose columns are orthonormal basis of $\mathbb{L}_1$. Then for some matrix $\mathbf{S}(t)\in\mathbb{R}^{k\times N}$ we can write the lagged vectors and the expression of $\mathbf{A}(t)$ in (35) as

$$\mathbf{A}(t)=\hat{\mathbf{U}}_\perp^T\mathbf{f_1}(t-L+1:t)$$
$$=\hat{\mathbf{U}}_\perp^T\mathbf{U}_1\mathbf{S}(t),$$

for all $t\geq T_0+L$. Notice that $\|\mathbf{S}(t)\|_F=\|\mathbf{f_1}(t-L+1:t)\|_F$. Now we give a lower bound on $\min_{t\geq T_0+L}\|\mathbf{A}(t)\|_F^2$

$$\min_{t\geq T_0+L}\|\hat{\mathbf{U}}_\perp^T\mathbf{U}_1\mathbf{S}(t)\|_F^2=\min_{t\geq T_0+L}\left[\|\mathbf{U}_1\mathbf{S}(t)\|_F^2-\|\hat{\mathbf{U}}_0^T\mathbf{U}_1\mathbf{S}(t)\|_F^2\right]$$
$$\geq\min_{t\geq T_0+L}\left[\|\mathbf{U}_1\mathbf{S}(t)\|_F^2-\|\hat{\mathbf{U}}_0^T\mathbf{U}_1\|_{op}^2\|\mathbf{S}(t)\|_F^2\right]$$
$$=(1-\|\hat{\mathbf{U}}_0^T\mathbf{U}_1\|_{op}^2)\min_{t\geq T_0+L}\|\mathbf{f_1}(t-L+1:t)\|_F^2$$
$$\geq(1-\|\hat{\mathbf{U}}_0^T\mathbf{U}_1\|_{op})\min_{t\geq T_0+L}\lambda_1^2(\mathbf{f_1}(t-L+1:t))$$
$$=(1-\|\hat{\mathbf{U}}_0^T\mathbf{U}_1\|_{op})\lambda_1^{\min,1}\tag{37}$$

where the fourth line follows by the fact that $\|\hat{\mathbf{U}}_0^T\mathbf{U}_1\|_{op}\leq1$, and in the last line we use the analogous notation to that in (6) for $\mathbf{f_1}(\cdot)$.

Let $\hat{\delta}:=\|\hat{\mathbf{U}}_0^T\mathbf{U}_1\|_{op}$, note that $\hat{\delta}$ describes the similarity between the post-change subspace and the *estimated* pre-change subspace. Recall that $\delta$ denotes the similarity between the post-change subspace and the *true* pre-change subspace, and that $\epsilon$ denotes the distance between the true pre-change subspace and its estimation. Now we aim to show that $\hat{\delta}\leq\delta+\epsilon$. To do so let us define the projection matrices as $\mathbf{\Pi}_0=\mathbf{U}_0\mathbf{U}_0^T$ and $\mathbf{\Pi}_\perp=\mathbf{U}_\perp\mathbf{U}_\perp^T$, where the columns of $\mathbf{U}_0$ and $\mathbf{U}_\perp$ are orthonormal basis of the pre-change subspace $\mathbb{L}_0$ and its orthogonal complement $\mathbb{L}_\perp$, respectively. Using the projection matrices we can write $\hat{\mathbf{U}}_0=\mathbf{\Pi}_0\hat{\mathbf{U}}_0+\mathbf{\Pi}_\perp\hat{\mathbf{U}}_0$. Then using the definition of $\hat{\delta}$

$$\hat{\delta}=\|(\mathbf{\Pi}_0\hat{\mathbf{U}}_0+\mathbf{\Pi}_\perp\hat{\mathbf{U}}_0)^T\mathbf{U}_1\|_{op}$$
$$\leq\|(\mathbf{\Pi}_0\hat{\mathbf{U}}_0)^T\mathbf{U}_1\|_{op}+\|(\mathbf{\Pi}_\perp\hat{\mathbf{U}}_0)^T\mathbf{U}_1\|_{op}$$
$$=\|\hat{\mathbf{U}}_0^T\mathbf{U}_0\mathbf{U}_0^T\mathbf{U}_1\|_{op}+\|\hat{\mathbf{U}}_0^T\mathbf{U}_\perp\mathbf{U}_\perp^T\mathbf{U}_1\|_{op}$$
$$\leq\|\hat{\mathbf{U}}_0^T\mathbf{U}_0\|_{op}\|\mathbf{U}_0^T\mathbf{U}_1\|_{op}+\|\hat{\mathbf{U}}_0^T\mathbf{U}_\perp\|_{op}\|\mathbf{U}_\perp^T\mathbf{U}_1\|_{op}$$
$$\leq\delta+\epsilon,\tag{38}$$

where the last inequality follows because $\|\hat{\mathbf{U}}_0^T\mathbf{U}_0\|_{op}$ and $\|\mathbf{U}_\perp^T\mathbf{U}_1\|_{op}\leq1$. Recall from Proposition 4.1 the definition of the event $\mathcal{E}=\{\epsilon<q\}$. Using this and plugging (38) in (37) we get

$$\min_{t\geq T_0+L}\|\hat{\mathbf{U}}_\perp^T\mathbf{U}_1\mathbf{S}(t)\|_F^2\geq(1-\delta-q)\lambda_1^{\min,1}.\tag{39}$$

Thus, by plugging the bound in (39) in the expression of $\mathbb{E}[D(t)]$ in (25), we get

$$\mathbb{E}[D(t)\,|\,\mathcal{E}]>N(L-k)\sigma^2+(1-\delta-q)\lambda_1^{\min,1}-c.\tag{40}$$

So any choice of $c$ as described in the theorem statement will make $\mathbb{E}[D(t)\,|\,\mathcal{E}]>0\ \forall t\geq T_0+L$. Let us define this positive quantity as:

$$\omega:=N(L-k)\sigma^2+(1-\delta-q)\lambda_1^{\min,1}-c.\tag{41}$$

**Step III. Bound on the tail probability of the CUSUM statistic.** Define the notation of the probability $P_{T_1}(\cdot):=P(\cdot\,|\,\tau=T_1,\mathcal{E})$. Recall from Definition 3.1, and for any $t>T_0$

$$y(t)=\max_{T_0<j\leq t}\left(\sum_{i=j}^t D(i)\right).$$

Let us now bound $\max_{t \geq T_0+L} \nu_t$ as follows:

$$\max_{t \geq T_0+L} \nu_t = \max_{t \geq T_0+L} 16\sigma^2 N\sqrt{L-k} + 2\sigma \sum_{n \in [N]} \|\mathbf{A}_n(t)\|_2$$

$$= 16\sigma^2 N\sqrt{L-k} + 2\sigma \max_{t \geq T_0+L} \sum_{n \in [N]} \|\hat{\mathbf{U}}_\perp^T f_{1,n}(t-L+1:t)\|_2$$

$$\leq 16\sigma^2 N\sqrt{L-k} + 2\sigma\sqrt{L} N R_1 \Gamma_{\alpha 1} \Gamma_{W1}$$

$$\leq 2\sigma N\sqrt{L}(\sigma + R_1 \Gamma_{\alpha 1} \Gamma_{W1}) =: \nu^*$$

Where $R_1, \Gamma_{\alpha 1}$ and $\Gamma_{W1}$ are properties of the function $\mathbf{f_1}(\cdot)$ defined as in Properties 2.1 and 2.2. Notice that $\nu^*$ is a valid parameter for each of the sub-exponential random variables $D'(t)$ for $t \geq T_0+L$. With this we can write

$$P_{T_1}(y(t) \leq h) \leq P_{T_1}\left(\max_{T_0+L \leq j \leq t}\left(\sum_{i=j}^{t} D(i)\right) \leq h\right)$$

$$\leq P_{T_1}\left(\sum_{i=T_0+L}^{t} D(i) \leq h\right)$$

$$= P_{T_1}\left(\sum_{\ell=1}^{L} \sum_{j=0}^{I(\ell)-1} D(T_0+L-1+j\times L+\ell) \leq h\right).$$

In the last equality, we grouped the observations into $L$ groups of independent detection scores, where $I(\ell)$ for $\ell \in [L]$ denotes the number of observations in the $\ell$-th group. Let the shifted index $t_0 := t-(T_0+L)$ and let $t_L := \lfloor t_0/L \rfloor$, then $I(\ell)$ is defined as follows:

$$I(\ell) = \begin{cases} t_L+1 & \text{if } \ell \leq (t_0+1) \pmod{L} \\ t_L & \text{if } \ell > (t_0+1) \pmod{L}. \end{cases}$$

Note that the event $\sum_{i=T_0+L}^{t} D(i) \leq h$ implies at least one of the events $\sum_{j=0}^{I(\ell)-1} D(T_0+L-1+j\times L+\ell) \leq h/L$ for some $\ell \in [L]$. That is,

$$P_{T_1}(y(t) \leq h) \leq P_{T_1}\left(\bigcup_{\ell=1}^{L}\left(\sum_{j=0}^{I(\ell)-1} D(T_0+L-1+j\times L+\ell) \leq h/L\right)\right)$$

$$\leq \sum_{\ell=1}^{L} P_{T_1}\left(\sum_{j=0}^{I(\ell)-1}\left(D'(T_0+L-1+j\times L+\ell) + \mathbb{E}[D(T_0+L-1+j\times L+\ell)\,|\,\mathcal{E}]\right) \leq h/L\right)$$

$$= \sum_{\ell=1}^{L} P_{T_1}\left(\sum_{j=0}^{I(\ell)-1} D'(T_0+L-1+j\times L+\ell) \leq h/L - \sum_{j=0}^{I(\ell)-1} \mathbb{E}[D(T_0+L-1+j\times L+\ell)\,|\,\mathcal{E}]\right)$$

$$\leq \sum_{\ell=1}^{L} P_{T_1}\left(\sum_{j=0}^{I(\ell)-1} D'(T_0+L-1+j\times L+\ell) \leq \frac{h}{L} - I(\ell)\omega\right)$$

$$\leq \sum_{\ell=1}^{(t_0+1)\pmod{L}} \exp\left(\frac{-1}{2}\left(\frac{((t_L+1)\omega - h/L)^2}{(t_L+1)\nu^{*2}} \wedge \frac{((t_L+1)\omega - h/L)}{\nu^*}\right)\right)$$

$$+ \sum_{\ell=(t_0+1)\pmod{L}+1}^{L} \exp\left(\frac{-1}{2}\left(\frac{(t_L\omega - h/L)^2}{t_L\nu^{*2}} \wedge \frac{t_L\omega - h/L}{\nu^*}\right)\right)$$

$$\leq L\exp\left(\frac{-1}{2}\left(\frac{(t_L\omega - h/L)^2}{(t_L+1)\nu^{*2}} \wedge \frac{t_L\omega - h/L}{\nu^*}\right)\right)$$

where the second line follows from the union bound and (36), the fourth line from the fact the $\mathbb{E}[D(t) \mid \mathcal{E}] > \omega \; \forall t > T_0 + L$, and the fifth line by direct application of Bernstein's inequality (helper Lemma B.7). Finally, for $t_0 \geq L$, i.e., $t \geq T_0 + 2L$, we have $t_L \geq 1$ and we can write following inequality:

$$P_{T_1}(y(t) \leq h) \leq L \exp\left(\frac{-1}{2}\left(\frac{(t_L\omega - h/L)^2}{(t_L+1)\nu^{*2}} \wedge \frac{t_L\omega - h/L}{\nu^*}\right)\right)$$

$$\leq L \exp\left(\frac{-1}{4L}\left(\frac{(t_0\omega - 2h)^2}{4t_0\nu^{*2}} \wedge \frac{t_0\omega - 2h}{\nu^*}\right)\right) \tag{42}$$

**Step IV. Bound on the** $ARL_1$ For any time index $T' > 0$, and conditioned on $\mathcal{E}$, we can express the $ARL_1$ as

$$ARL_1 = E_{T_1}[\hat{\tau} - T_0 \mid \tau = T_1, \mathcal{E}]$$

$$= \sum_{t'=1}^{\infty} t' P_{T_1}(\hat{\tau} - T_0 = t')$$

$$= \sum_{t'=1}^{T'+L} t' P_{T_1}(\hat{\tau} - T_0 = t') + \sum_{t'=T'+L+1}^{\infty} t' P_{T_1}(\hat{\tau} - T_0 = t'). \tag{43}$$

The first term on the right hand side of (43) can be bounded as

$$\sum_{t'=1}^{T'+L} t' P_{T_1}(\hat{\tau} - T_0 = t') \leq \sum_{t'=1}^{T'+L} (T'+L) P_{T_1}(\hat{\tau} - T_0 = t')$$

$$\leq T' + L. \tag{44}$$

The second term in the right hand side of (43) can be bounded as

$$\sum_{t'=T'+L+1}^{\infty} t' P_{T_1}(\hat{\tau} - T_0 = t') = (T'+L+1)P_{T_1}(\hat{\tau} - T_0 = T'+L+1) + (T'+L+2)P_{T_1}(\hat{\tau} - T_0 = T'+L+2) + \ldots$$

$$= (T'+L)P_{T_1}(\hat{\tau} - T_0 \geq T'+L+1) + \sum_{t'=T'+L+1}^{\infty} P_{T_1}(\hat{\tau} - T_0 \geq t')$$

$$\leq (T'+L)P_{T_1}(y(T_0+T'+L) \leq h) + \sum_{t'=T'+L}^{\infty} P_{T_1}(y(T_0+t') \leq h) \tag{45}$$

Where the last line follows from the definition of $\hat{\tau}$ which indicates that $P_{T_1}(\hat{\tau} - T_0 > t') \leq P_{T_1}(y(T_0+t') \leq h)$. By plugging (44) and (45) back in (43) we get

$$ARL_1 \leq T' + L + (T'+L)P_{T_1}(y(T_0+T'+L) \leq h) + \sum_{t'=T'+L}^{\infty} P_{T_1}(y(T_0+t') \leq h)$$

$$= (T'+L)(1 + P_{T_1}(y(T_0+T'+L)) \leq h)) + \sum_{t'=T'+L}^{\infty} P_{T_1}(y(T_0+t') \leq h). \tag{46}$$

**Step V. Choice of** $T'$. Since (46) is valid for any choice of $T' > L$, let us pick a value of the time index as

$$T' := \left\lceil \frac{2h}{\omega}\left(\frac{\omega + 2\nu^*}{\omega + \nu^*}\right)\right\rceil + L - 1. \tag{47}$$

We can show that for any $t_0 \geq T'$ i.e. $t \geq T' + T_0 + L$,

$$\frac{(t_0\omega - 2h)^2}{4t\nu^{*2}} = \frac{t_0\omega - 2h}{4t_0\nu^*}\frac{t_0\omega - 2h}{\nu^*}$$

$$\geq \frac{T'\omega - 2h}{4T'\nu^*} \frac{t_0\omega - 2h}{\nu^*}$$

$$\geq \frac{\frac{2h}{\omega}\left(\frac{\omega+2\nu^*}{\omega+\nu^*}\right)\omega - 2h}{4\frac{2h}{\omega}\left(\frac{\omega+2\nu^*}{\omega+\nu^*}\right)\nu^*} \frac{t_0\omega - 2h}{\nu^*}$$

$$= \frac{\omega}{4\omega + 8\nu^*} \frac{t_0\omega - 2h}{\nu^*}.$$

Notice that $\frac{\omega}{4\omega+8\nu^*} < 1$, thus $\frac{\omega}{4\omega+8\nu^*}\frac{t\omega-2h}{\nu^*} \leq \frac{t_0\omega - h}{\nu^*}$. Recall that $t_0 := t - (T_0 + L)$ and thus, using both inequalities, and the Bernstein bound in (42) we have:

$$P_{T_1}(y(t) \leq h) \leq L \exp\left(\frac{-1}{4L}\left(\frac{((t-(T_0+L))\omega - 2h)^2}{4t_0\nu^{*2}} \wedge \frac{(t-(T_0+L))\omega - 2h}{\nu^*}\right)\right)$$

$$\leq L \exp\left(\frac{-1}{16L}\left(\frac{\omega}{\omega+2\nu^*}\frac{(t-(T_0+L))\omega - 2h}{\nu^*}\right)\right). \tag{48}$$

Using this and the choice of $T'$ in (47) we can give bounds on each term in (46). For the first and second term, we have:

$$T' \leq \frac{2h}{\omega}\left(\frac{\omega+2\nu^*}{\omega+\nu^*}\right) + L \tag{49}$$

$$P_{T_1}(y(T'+T_0+L) \leq h) \leq L \exp\left(\frac{-1}{16L}\left(\frac{\omega}{\omega+2\nu^*}\frac{T'\omega - 2h}{\nu^*}\right)\right)$$

$$\leq L \exp\left(\frac{-1}{16L}\left(\frac{\omega}{\omega+2\nu^*}\frac{\frac{2h}{\omega}\left(\frac{\omega+2\nu^*}{\omega+\nu^*}\right)\omega - 2h}{\nu^*}\right)\right)$$

$$= L \exp\left(\frac{-2\omega h}{16L(\omega+2\nu^*)(\omega+\nu^*)}\right)$$

$$\leq L \exp\left(\frac{-2\omega h}{16L(\omega+2\nu^*)^2}\right) \tag{50}$$

For the third term, we have:

$$\sum_{t'=T'+L}^{\infty} P_{T_1}(y(T_0+t') \leq h) \leq L \sum_{t'=T'+L}^{\infty} \exp\left(\frac{-1}{16L}\left(\frac{\omega}{\omega+2\nu^*}\frac{(T_0+t'-T_0-L)\omega - 2h}{\nu^*}\right)\right)$$

$$= L \sum_{t'=T'+L}^{\infty} \exp\left(\frac{-1}{16L}\left(\frac{\omega}{\omega+2\nu^*}\frac{(t'-L)\omega - 2h}{\nu^*}\right)\right)$$

$$= L \sum_{t'=T'+L}^{\infty} \exp\left(\frac{-1}{16L}\left(\frac{\omega}{\omega+2\nu^*}\frac{(t'-L-T')\omega + T'\omega - 2h}{\nu^*}\right)\right)$$

$$\leq L \sum_{t'=T'+L}^{\infty} \exp\left(\frac{-2\omega h}{16L(\omega+2\nu^*)^2} + \frac{-\omega^2(t'-L-T')}{16L\nu^*(\omega+2\nu^*)}\right)$$

$$= L \exp\left(\frac{-2\omega h}{16L(\omega+2\nu^*)^2}\right) \sum_{t'=T'+L}^{\infty} \exp\left(\frac{-\omega^2}{16L\nu^*(\omega+2\nu^*)}\right)^{t'-L-T'}$$

$$= L \exp\left(\frac{-2\omega h}{16L(\omega+2\nu^*)^2}\right) \sum_{t''=0}^{\infty} \exp\left(\frac{-\omega^2}{16L\nu^*(\omega+2\nu^*)}\right)^{t''}$$

$$= L \exp\left(\frac{-2\omega h}{16L(\omega+2\nu^*)^2}\right) \frac{1}{1-\exp\left(\frac{-\omega^2}{16L\nu^*(\omega+2\nu^*)}\right)}$$

$$\leq \frac{L\exp\left(\frac{-2\omega h}{16L(\omega+2\nu^*)^2}\right)}{1-\exp\left(\frac{-\omega^2}{2\nu^*(\omega+16L\nu^*)}\right)} \tag{51}$$

Bringing all these terms together, we have the following bound on the ARL1,

$$ARL_1 \leq \left(\frac{2h}{\omega}\left(\frac{\omega+2\nu^*}{\omega+\nu^*}\right)+2L\right)\left(1+L\exp\left(\frac{-2\omega h}{16L(\omega+2\nu^*)^2}\right)\right)+\frac{L\exp\left(\frac{-2\omega h}{16L(\omega+2\nu^*)^2}\right)}{1-\exp\left(\frac{-\omega^2}{16L\nu^*(\omega+2\nu^*)}\right)} \tag{52}$$

$$\leq 2L+\frac{4hL}{\omega}\left(\frac{\omega+2\nu^*}{\omega+\nu^*}\right)+L\exp\left(\frac{-\omega h}{16L(\omega+2\nu^*)^2}\right)\left(2L+\frac{1}{1-\exp\left(\frac{-\omega^2}{16L\nu^*(\omega+2\nu^*)}\right)}\right) \tag{53}$$

$$\leq C_0+C_1 h+C_2\exp(-C_3 h) \tag{54}$$

Where $C_0$, $C_1$, $C_2$, and $C_3$ are constants that depend only on $\sigma, c, N, L, R_1, \Gamma_{\alpha 1}, \Gamma_{W1}$.

## B.5 Proof of Theorem 4.3

*Proof.* For a feasible choice of $c$ that satisfy the conditions (14) and (15), we need to select $c$ in the range

$$N(L-k)\sigma^2 + qR_0\lambda_1^{\max,0} < c < N(L-k)\sigma^2 + (1-\delta-q)\lambda_1^{\min,1}. \tag{55}$$

For this range to be non-empty it is required that

$$qR_0\lambda_1^{\max,0} < (1-\delta-q)\lambda_1^{\min,1} \tag{56}$$

or equivalently

$$\delta < 1 - \left(1 + R_0\frac{\lambda_1^{\max,0}}{\lambda_1^{\min,1}}\right)q$$

Which completes the proof. $\qquad\square$

## C  Algorithms Pseudocode

In this section, we provide the pseudocode for the two variants of our proposed algorithm.

---

**Algorithm 1:** mSSA Change Point Detection Algorithm

---

**Parameters:** $T_0$, $L$, $\hat{k}$, $c$, $h$
**Data:** Multivariate time series $\mathbf{X}(t) \in \mathbb{R}^N$
**Result:** Estimated change point $\hat{\tau}$
**Initialize:** $y(T_0) = 0$
Use the segment $\mathbf{X}(1:T_0)$ to construct the stacked Page matrix $\mathbf{Z_X}$ as in (4);
Compute the SVD of $\mathbf{Z_X}$ to get the left singular vectors $\mathbf{u}_i \ \forall i \in [L]$;
Construct the matrix $\hat{\mathbf{U}}_\perp = [\mathbf{u}_{\hat{k}+1},...,\mathbf{u}_L]$;
**for** $t \leftarrow T_0+1, T_0+2, T_0+3...,$ **do**
    Construct the matrix $\mathbf{X}(t-L+1:t)$ as in (5);
    Compute $D(t)$ as in (10);
    Compute $y(t)$ as in (8);
    **if** $y(t) \geq h$ **then**
        $\hat{\tau} = t$;
        **Break**;
    **end**
**end**

---

---

**Algorithm 2:** mSSA-MW Change Point Detection Algorithm

---

**Parameters:** $T_0$, $L$, $\hat{k}$, $c$, $h$
**Data:** Multivariate time series $\mathbf{X}(t) \in \mathbb{R}^N$
**Result:** Estimated change point $\hat{\tau}$
**Initialize:** $y(T_0) = 0$
$T_L \leftarrow T_0 + L$;
**for** $t \leftarrow T_L, T_L+1, T_L+2,...,$ **do**
    Use the segment $\mathbf{X}(t-T_L+1:t-L)$ to construct the stacked Page matrix $\mathbf{Z_X}$ as in (4);
    Compute the SVD of $\mathbf{Z_X}$ to get the left singular vectors $\mathbf{u}_i \ \forall i \in [L]$;
    Construct the matrix $\hat{\mathbf{U}}_\perp = [\mathbf{u}_{\hat{k}+1},...,\mathbf{u}_L]$;
    Construct the matrix $\mathbf{X}(t-L+1:t)$ as in (5);
    Compute $D(t)$ as in (10);
    Compute $y(t)$ as in (8);
    **if** $y(t) \geq h$ **then**
        $\hat{\tau} = t$;
        **Break**;
    **end**
**end**

---

# D Experiment Details

## D.1 Parameter Configuration

In this section, we give details of the implementation and parameter configurations used for each of the algorithms in the comparative experiments in Sections 5 and 6. We provide the following statistics about each dataset to be utilized by all algorithms (if applicable): (1) total number of change points in the dataset $T_C$ (2) number of time series in the dataset $TS$ (3) average interval length between two consecutive change points in the entire dataset $I$. Note that in real-world applications, such knowledge can be obtained from historical data or knowledge of the nature of the change points.

**mSSA and mSSA-MW.**

- **$T_0$**: we set $T_0 = 0.6 \times I$.
- **L**: we select $L = \lfloor \kappa_L \times \sqrt{T_0} \rfloor$ where we consider $\kappa_L = 1$ (default),0.7, and 0.3.
- **$\hat{k}$**: We follow the thresholding rule

$$\hat{k} = \min\{i | \sum_{j=1}^{i} \lambda_j^2(\mathbf{Z_X}) \geq \kappa_k \sum_{j=1}^{L} \lambda_j^2(\mathbf{Z_X})\}, \tag{57}$$

  where $\lambda_j(\mathbf{Z_X})$ is the $j^{th}$ largest singular value of $\mathbf{Z_X}$, and $\kappa_k < 1$ is a constant. We consider $\kappa_k = 0.95$ (default), and 0.5, and we also consider constant low orders of $\hat{k} = 3$ and 5.

- **c**: the choice of the parameter $c$ must be done such that the detection score satisfies Property 3.1. Concretely, it requires selecting $c > 0$ such that $N(L-k)\sigma^2 + \|\mathbf{A}(t)\|_F^2 - c < 0$ for all $t < \tau$ where $\mathbf{A}(t) :=$ $\hat{\mathbf{U}}_\perp^T \mathbf{f_0}(t-L+1:t)$. To do so, we first estimate $\sigma^2$ (as $\hat{\sigma}^2$) from the noise component of the base matrix obtained in the thresholding step of mSSA. Secondly, we estimate the maximum distance $\max_{t<\tau} \|\mathbf{A}(t)\|_F^2$ (as $\|\hat{\mathbf{A}}(t)\|_{F,\max}^2$) by cross validation. That is, we use 90% of the L-lagged vectors in the base matrix to estimate the pre-change subspace and we compute the detection score for the remaining 10% of the L-lagged vectors and take the maximum. Finally, we use our estimation of the time series order $\hat{k}$ and set $c$ to be slightly larger than $N(L-\hat{k})\hat{\sigma}^2 + \|\hat{\mathbf{A}}(t)\|_{F,\max}^2$.

- **h**: the choice of the detection threshold is also based on the estimation of the maximum distance $\max_{t<\tau} \|\mathbf{A}(t)\|_F^2$. We take $h = \kappa_h \|\hat{\mathbf{A}}(t)\|_{F,\max}^2$ where we consider $\kappa_h = 1,5$ (default), and 10.

**BinSeg.** We use the implementation in the R package *changepoint* [21] and consider the following parameter variations in the grid search:

- **Function:** cpt.mean (default), cpt.var, cpt.meanvar.
- **Penalty:** None, SIC, MBIC (default), AIC, Hannan-Quinn, Asymptotic (with p-value = 0.05).
- **Test Statistic:** Normal (default), CUSUM, CSS.
- **Maximum number of CP:** $T_C$, $T_C/TS$ (defulat).

**Microsoft SSA.** We use the implementation in the Python module *NimbusML* which provides access to the *ML.NET* framework [29]. We consider the following parameter variations in the grid search:

- **Training window size:** $0.6 \times I$.
- **Seasonal window size:** $\lfloor \sqrt{0.6 \times I} \rfloor$ (default), $\lfloor 0.7\sqrt{0.6 \times I} \rfloor$, $\lfloor 0.3\sqrt{0.6 \times I} \rfloor$.
- **Change history length:** 10 (default), 30 , 50
- **Error function:** SignedDifference (default), AbsoluteDifference, SignedProportion, AbsoluteProportion, SquaredDifference.
- **Martingale:** Power (default), Mixture
- **$\epsilon$ (for power martingale):** 0.1 (default), 0.5
- **Confidence:** 95.0

**KL-CPD.** We use the Python implementation provided by the authors of the paper [7]. We follow the authors' recommendation for setting the parameters and consider the following parameter variations in the grid search:

- **Real MMD loss coefficient:** 0.001, 0.1 (default),1 ,10
- **Reconstruction loss coefficient:** 0.001 (default), 0.1, 1, 10

- **Window size:** 25

**BOCPDMS.** We use the Python implementation provided by the authors of the paper [23] and consider the following parameter variations in the grid search:

- `intensity`: 50, 100 (default), 200
- `prior_on_a`: 0.01, 1.0 (default), 100
- `Prior_on_b`: 0.01, 1.0(default), 100

## D.2   Benchmark Datasets

In this section we give details of the benchmark datasets used in the experiments in Section 5.

**Beedance.**[6] The trajectory of bee movements while it performs what is known is the waggle dance. The three dimensions of the time series represent the pixel locations in $x$ and $y$ axes, and the change in angle extracted from a video clip of the moving bee. The change points represent switching between three states in the bee dance: left turn, right turn, and waggle. The dataset is generated by [33] and it has been considered in other CPD work such as [7, 37, 23].

**HASC.**[7] Human activity data collected using a portable three-axis accelerometer. The three dimensions of the time series represent the acceleration recorded by the device along the $x$, $y$, and $z$ axes. The change points represent switching between six activities: staying, walking, jogging, skipping, climbing up the stairs, and climbing down the stairs. This is a sample dataset provided by the 2011 Human Activity Sensing Consortium challenge. This dataset has been considered in other CPD work such as [7, 27, 3].

**Occupancy.**[8] This dataset is used for the task of detecting the occupancy of office space using various measurements of the room condition. The four dimensions of the time series represent the temperature, relative humidity, light intensity, and $CO_2$ levels in the room. The change points represent changes in the occupancy level in the room. The dataset is generated by [6] and it has been considered in other CPD work such as [42].

**Yahoo.**[9] This is the A4Benchmark dataset provided as part of Yahoo's S5 datasets for anomaly detection. It contains univariate synthetic time series with artificially introduced change points. This dataset is commonly used as a benchmark in time series anomaly detection [15].

## D.3   Synthetic Data Generation

To generate a time series $\mathbf{X}(t) \in \mathbb{R}^N$ with $t \in [T]$, we start by generating $R \leq N$ fundamental time series $W_r(t)$, $r \in [R]$. Each fundamental time series is a mixture of $H$ harmonics and polynomial trend component and is generated as

$$W_r(t) = t^{\alpha_r} + \sum_{i=1}^{H} \beta_{r,i} \sin\left(\frac{\omega_{r,i} t}{T}\right) + C_r, \tag{58}$$

with parameters $\alpha_r$, $\beta_{r,i}$, $\omega_{r,i}$, and $C_r$ for $r \in [R]$ ans $i \in [H]$. Each latent time series $f_n(t)$ for $n \in [N]$ is then constructed as a linear combination of the fundamental time series

$$f_n(t) = \mathbf{V_n}^T \mathbf{W}(t), \tag{59}$$

where $\mathbf{W}(t) := [W_1(t), ..., W_R(t)]$, and $\mathbf{V_n} \in \mathbb{R}^R$ is a random vector whose components are selected form $U(0,1)$. Finally, the observations $X_n(t)$ are generated by adding i.i.d. Gaussian noise with zero mean and variance $\sigma^2$. Change points are artificially introduced into the data by changing one or more of the parameters in the construction of the fundamental time series. Using this data generation process, we construct the following four datasets:

**Jumping mean.** We generate 20 time series for this dataset. Each time series is univariate ($N = 1$, $R = 1$) with length $T = 5000$. A change point is inserted every 1000 time steps by changing the value of the parameter $C_1$. The rest of the parameters are kept constant across the time series. The time series is generated as the sum of three harmonics ($H = 3$) and no trend component ($\alpha_1 = 0$). The noise is distributed as $e_1(t) \sim N(0, 0.01)$. The rest of the parameters are selected randomly from the following ranges:

- $\omega_{1,i} \sim U(5, 105)$
- $\beta_{1,i} \sim U(0, 7)$

---

[6]Obtained from: `https://www.cc.gatech.edu/~borg/ijcv_psslds/`
[7]Obtained from: `http://hasc.jp/hc2011/download.html`
[8]Obtained from: `https://archive.ics.uci.edu/ml/datasets/Occupancy+Detection+/`
[9]Obtained from: `https://webscope.sandbox.yahoo.com/catalog.php?datatype=s&did=70`

- $C_1 \sim U(0,10)$ (varying parameter)

**Scaling signal energy.** We generate 20 time series for this dataset. Each time series in this dataset is univariate ($N=1$, $R=1$) with length $T=5000$. A change point is inserted every 1000 time steps by changing the values of the parameters $\beta_{1,i}$ for $i \in [H]$. The rest of the parameters are kept constant across the time series. The time series is generated as the sum of three harmonics ($H=3$) and no trend component ($\alpha_1 = 0$). The noise is distributed as $e_1(t) \sim N(0,0.01)$. The rest of the parameters are selected randomly from the following ranges:

- $\omega_{1,i} \sim U(5,105)$
- $\beta_{1,i} \sim U(0,u)$ where $u$ alternates between 5 and 7 after each change point (varying parameter)
- $C_1 = 0$

**Changing frequency.** We generate 20 time series for this dataset. Each time series is univariate ($N=1$, $R=1$) with length $T=5000$. A change point is inserted every 1000 time steps by changing the values of the parameters $\omega_{1,i}$ for $i \in [H]$. The rest of the parameters are kept constant across the time series. The time series is generated as the sum of three harmonics ($H=3$) and no trend component ($\alpha_1 = 0$). The noise is distributed as $e_1(t) \sim N(0,0.01)$. The rest of the parameters are selected randomly from the following ranges:

- $\omega_{1,i} \sim U(5,u)$ where $u$ alternates between 35 and 75 after each change point (varying parameter)
- $\beta_{1,i} \sim U(0,7)$
- $C_1 = 0$

**Mixed changes.** We generate 5 time series for this dataset. Each time series in this dataset is multivariate ($N=100$, $R=10$) with length $T=1000$. A change point is inserted every 200 time steps by changing the values for the parameters $\omega_{r,i}$ and $\beta_{r,i}$ for $r \in [R]$, $i \in [H]$. The parameters $C_r$ and $\alpha_r$ are kept constant across the time series. The time seires is generated as the sum of three harmonics ($H=3$) and linear trend component ($\alpha_r = 1 \ \forall r \in [10]$). The noise is distributed as $e_n(t) \sim N(0,49) \ \forall n \in [100]$. The rest of the parameters are selected randomly from the following ranges:

- $\omega_{1,i} \sim U(5,u)$ where $u$ alternate between 35 and 75 after each change point (varying parameter)
- $\beta_{1,i} \sim U(0,7)$ (varying parameter)
- $C_r = 0$

### D.4 Multivariate Data Helps: Experiments Details

In this experiments, we start by constructing ten fundamental time series, as described in (58). The time series are generated as the sum of three harmonics ($H=3$) and no trend component ($\alpha_r = 0 \ \forall r \in [10]$). The noise is distributed as $e_n(t) \sim N(0,0.5) \ \forall n \in [N]$. The rest of the parameters are selected randomly from the following ranges:

- $\omega_{1,i} \sim U(5,u)$ where $u$ is 3005 before the change point and 4005 after the change point (changing parameter)
- $\beta_{1,i} \sim U(0,1)$ (changing parameter)
- $C_r = 0$

We then construct each of the 25 latent time series as described in (59). In each of the 10 trial, the latent time series remain the same and only the noise is re-sampled in each trial.

The parameters used in this algorithm are $T_0 = 800$, $L = \sqrt{N \times T_0}$, and $\hat{k}$ is selected according the rule in (57) with $\kappa_k = 0.95$. The thresholds $h$ are chosen between the $40^{th}$ and $100^{th}$ quantiles of the CUSUM score in the no change run.

### D.5 CPD in Multivariate Time Series with Univariate Methods

Recall that the BinSeg and Microsoft SSA methods only support univariate time series. In Section 5, we report their performance on multivariate datasets by reducing the data to univariate time series through taking the L2 norm of the the multidimensional observations. Here, we explore two other approaches for detecting changes in multivariate time series using univariate methods.

- **Majority vote:** in this approach, we run the CPD algorithm on each individual time series (i.e., dimensions). Then, we take a majority vote (among the different dimensions) to decide whether a point is designated as a change point or not. More precisely, a true detection is counted if the algorithm detects a change point, within a margin of $\eta = 10$ points from the labeled change point, in half or more of the individual time series.

- **At least one:** in this approach, we again run the CPD algorithm on each individual time series, but we consider all change points triggered by the algorithm across all time series. More precisely, a true detection is counted if the algorithm detects a change point within a margin of $\eta = 10$ points from the labeled change point in *any* of the individual time series, without double counting.

In Table 4, we compare the performance of the BinSeg and Microsoft SSA algorithms using these two detection approaches to the norm approach presented in Section 5. We find that taking the norm is better than the majority vote approach, while the at least one approach is slightly better. We note that our proposed mSSA method outperforms the best of the three.

Table 4: Mean of F1-scores for different approaches of applying the BinSeg and Microsoft SSA algorithms to multivariate time series.

| | **Real-world Datasets** | | | | | |
| | Beedance | | HASC | | Occupancy | |
| | Default | Best | Default | Best | Default | Best |
| --- | --- | --- | --- | --- | --- | --- |
| BinSeg (norm) | 0.597 | 0.097 | **0.304** | **0.161** | 0.308 | 0.308 |
| BinSeg (majority vote) | 0.371 | 0.096 | 0.192 | 0.023 | 0.143 | 0.050 |
| BinSeg (at least one) | **0.629** | **0.109** | 0.295 | 0.151 | **0.485** | **0.340** |
| Microsoft SSA (norm) | 0.583 | 0.279 | 0.265 | 0.049 | **0.462** | **0.375** |
| Microsoft SSA (majority vote) | 0.462 | 0.095 | 0.179 | **0.151** | 0.148 | 0.075 |
| Microsoft SSA (at least one) | **0.619** | **0.338** | **0.280** | 0.043 | 0.286 | 0.113 |