# OpenReview forum: "Change Point Detection via Multivariate Singular Spectrum Analysis"
_NeurIPS.cc/2021/Conference — NeurIPS 2021 Poster_

### Official Review · Reviewer_xiRb · 2021-07-07

**Rating:** 6
**Confidence:** 4

**Summary:**

The paper studies change-point detection in time-series using a combination multi-variate singular spectrum analysis (SSA) and CUSUM.   While multi-variate SSA and CUSUM are established techniques, the contribution of the paper is to (a) analyze the online CPD setting with a dependent (non-iid) timeseries. (b) Conduct rigorous analysis of the tradeoff of detection delay vs. false-alarm within the MSSA model.  (c) The exact version of MSSA is a bit different from the classical one.

**Ethical Concerns:**

N/A.

**Limitations And Societal Impact:**

I'm not aware of societal impact of this work.

**Main Review:**

The papers takes a fresh look at applying classical techniques MSSA and CUSUM to change-point detection.   While the combination of SSA and CUSUM has already been proposed for CPD (as cited in ref [30], Moskvina and Zhigljavsky, 2003),  and MSSA itself has been proposed a decade ago (e.g. 2013 -- also Zhigljavsky et al) -- the main contribution of this paper is a formal theoretical analysis, and a detailed numerical evaluation.

The paper is mostly clearly written (some caveats below), and I believe the style of analysis (using concentration inequalities, and the distribution of subspace angles and matrix singular values) is a contribution to the SSA / CPD literature.  I would prefer the paper to be more upfront about what's new and what has been done before, and novelty:  in the appendix there's a good description of the existing literature, and the contribution that this paper makes.  I'd suggest moving a few sentences into the main paper.  The main paper itself is a bit cryptic and lacks references (e.g.  ref for early MSSA would be good).

I'm not clear that the evaluation is as strong as it could be -- how were the baselines chosen?  Is using the norm of multivariate time-series and furnishing to univariate CPD a good baseline?  Can you maybe run univariate CPD on each time-series  and do some sort of a voting -- that may have more of a chance at detecting richer classes of changes? Other than that, experiments seem to have been done thoughtfully.

The setting in the introduction could be made clearer (it's clear by the end of the paper -- but confusing at the start):
1) You mention f0 and f1 on page 2.   Are you detecting a change from f0 to f1 when both are known,  or do you only know f0, and declare a change when you're sufficiently far from f0?
2) Page 3 -- let H(t) -- be the random variable equal to the accepted hypothesis at time t.   This is too informal -- you're not 'accepting' a hypothesis H0 vs H1 -- but trying to reject H0?
3) The description of detecting slow-moving changes vs.  abrupt changes can be made clearer as well.  It seems that the shift-downward constant is related to the baseline slow-moving rate of change. Can you make this clear.
4) What is a 'valid' CUSUM detection score?  Is it bad to have an 'invalid' score? What does it mean?
5) I was happy to see that in addition to bounds on ARL0 and ARL1, the paper analyzes when these are not vacuous. Also happy to see various suggested heuristics choices for various parameters. Sign of practical mindset.
6) Why did you chose the particular 4 methods?  Are these representative of state-of-the-art?   Picking the norm of multivariate time-series and using univariate CPD methods seems like a rather suboptimal approach.
7) The tradeoff of ARL0 vs ARL1 changes dramatically w.r.t number of time-series.  Is the subspace distance as a function of N somhow normalized?  It sounds like higher dimensional problems are simply much easier
(subspace angle after the change is much larger in higher dimensions).

Additional comments:
[1] Minor:  notation Gamma^alpha -- this is just a constant, but suggests some power law w.r.t. alpha.  same for Gamma^w.
[2] You use [1] as the ref for MSSA -- how is it different from earlier MSSAs that you describe in appendix? (add a comment + ref in main paper)  .
[3] Property 2.2. The language makes it confusing as to whether it's an assumption or implication.
[4] For finite sums of harmonics, low-degree polys, and exp-functions Page matrix is low-rank.  Comment on rank vs. num terms?  In general for your data-sets in the experimental section, what rank did you end up choosing (for default / best params)?



**Time Spent Reviewing:**

4

---

> ### Author Response · Authors · 2021-08-10
> **Response**
>
> We thank the reviewer for highlighting the contribution of this work over previous SSA-based methods for change point detection. Below we answer the comments and questions raised by the reviewer.
>
> > I would prefer the paper to be more upfront about what's new and what has been done before, and novelty: in the appendix there's a good description of the existing literature, and the contribution that this paper makes. I'd suggest moving a few sentences into the main paper. The main paper itself is a bit cryptic and lacks references (e.g. ref for early MSSA would be good).
>
> We thank the reviewer for this comment and the characterization of the paper's contributions. We note that the mentioned references were placed in the appendix due to space constraints. In our revision of the paper, we will follow the reviewer's suggestion and cover the following points in the main body of the paper:
> - Referencing earlier work on SSA for change point detection.
> - Referencing early work on mSSA and describing the difference between the version we use and the original one.
> - Highlighting the contribution we made to this line of research in terms of (1) the theoretical analysis (2) empirical results (3) model modifications over original mSSA.
>
> > You mention f0 and f1 on page 2. Are you detecting a change from f0 to f1 when both are known, or do you only know f0, and declare a change when you're sufficiently far from f0?
>
> The latter description is more accurate. Specifically, while we impose certain assumptions on both $\mathbf{f}_0$ and $\mathbf{f}_1$, i.e. Property 2.1, and Property 2.2, we only have *noisy* information about $\mathbf{f}_0$.  That is,  when applying the change point detection algorithm, we estimate $\mathbf{f}_0$ from the noisy observations of the time series ($\mathbf{X}(t)$), while $\mathbf{f}_1$ is not known. As the reviewer pointed out, we declare a change point when the observations are sufficiently far from $\mathbf{f}_0$.
>
> > Page 3 -- let H(t) -- be the random variable equal to the accepted hypothesis at time t. This is too informal -- you're not 'accepting' a hypothesis H0 vs H1 -- but trying to reject H0?
>
> We thank the reviewer for the remark. We agree that the wording of this definition of $H(t)$ needs revisiting. More formally, we can define $H(t)$ as the indicator random variable $\mathbb{1}_{H_0 \text{ is rejected at time }t}$. Then the change point can be defined as $\inf$ \{ $t | H(t) = 1$\}
>
> > The description of detecting slow-moving changes vs. abrupt changes can be made clearer as well. It seems that the shift-downward constant is related to the baseline slow-moving rate of change. Can you make this clear.
>
> In this paper, we focus theoretically on abrupt changes, and we believe that addressing gradual changes is an interesting direction for future work.  For practical considerations, we refer the reviewer to our discussion of the two mSSA implementations: "fixed base matrix" and "moving window". Specifically, we believe that the fixed base matrix implementation is more sensitive to gradual changes as the subspace is constructed at the beginning of the process. While in the moving window implementation, the subspace is updated with every timestep, making it harder to capture small changes that accumulate over time.
>
> > What is a 'valid' CUSUM detection score? Is it bad to have an 'invalid' score? What does it mean?
>
> A valid detection score is one that satisfies Property 3.1. Notice that in Theorems 4.1 and 4.2, the bounds are derived with certain restrictions on the choice of the shift downwards constant $c$, which is a choice that makes property 3.1 hold. These results do not necessarily hold for invalid detection scores.
>
> > Why did you chose the particular 4 methods? Are these representative of state-of-the-art? Picking the norm of multivariate time-series and using univariate CPD methods seems like a rather suboptimal approach.
>
> > how were the baselines chosen? Is using the norm of multivariate time-series and furnishing to univariate CPD a good baseline? Can you maybe run univariate CPD on each time-series and do some sort of a voting -- that may have more of a chance at detecting richer classes of changes?
>
> The choice of the comparison methods was made with the following criteria in mind:
> - BinSeg is a classical method that proved to have superior performance according to [41]
> - BOCPDMS and KL-CPD are recent work that builds on a long line of research on Baysian and Kernel change point detection methods - two important classes of methods in the CPD literature.
> - Microsoft SSA is a commercial tool that uses the classical SSA method. We compare with it due to the maturity of the tool and to gauge how our method compares to the classical SSA variant.
>
> We have performed the experiments with running CPD on individual time series (i.e., dimensions) as suggested. This was done in two ways: (i) by taking a majority vote (i.e., a true detection is counted if the algorithm detects a change point, within a margin of 10 points from the labeled change point, in half or more of the individual time series); (ii) by considering all points triggered by the algorithm across all time series (i.e., a true detection is counted if the algorithm detects a change point in *any* of the individual time series, without double counting). Below we show the results for these two approaches.  We notice that taking the norm is better than the first approach, while the second approach is slightly better. We note that mSSA MW still outperforms the best of these three approaches.
>
> | Method                   | Beedance Best | Beedance Default | HACS Best | HASC Default | Occupancy Best | Occupance Default |
> |--------------------------|---------------|------------------|-----------|--------------|----------------|-------------------|
> | BinSeg (norm)            | 0.597         | 0.097            | 0.304     | 0.161        | 0.308          | 0.308             |
> | BinSeg (majority)        | 0.371         | 0.096            | 0.192     | 0.023        | 0.143          | 0.05              |
> | BinSeg (all cp)          | 0.629         | 0.109            | 0.295     | 0.151        | 0.485          | 0.34              |
> | Microsoft SSA (norm)     | 0.583         | 0.279            | 0.265     | 0.049        | 0.462          | 0.375             |
> | Microsoft SSA (majority) | 0.462         | 0.095            | 0.179     | 0.151        | 0.148          | 0.075             |
> | Microsoft SSA (all cp)   | 0.619         | 0.338            | 0.28      | 0.043        | 0.286          | 0.113             |
> | mSSA MW                  | 0.659         | 0.500            | 0.327     | 0.177        | 0.783          | 0.480             |
>
> > The tradeoff of ARL0 vs ARL1 changes dramatically w.r.t number of time-series. Is the subspace distance as a function of N somhow normalized? It sounds like higher dimensional problems are simply much easier (subspace angle after the change is much larger in higher dimensions).
>
> The detection score is not normalized as a function of N. The main explanation for the improvement in the tradeoff is that as we have more observations ($N\times T$), our estimation of the pre-change subspace becomes more accurate (refer to proposition 4.1 for formal quantification of how the subspace estimation error depends on the matrix size).  Please note that in this experiment, the order $k$ (see Definition 2.2) of the time series is fixed as we increase the number of dimensions.
>
> > Minor: notation Gamma^alpha -- this is just a constant, but suggests some power law w.r.t. alpha. same for Gamma^w
>
> Thank you for the note, Gammas are indeed constants, and the superscripts $\alpha$ and $W$ are only used to denote relation to $\alpha_{\cdot,r}$ and $W_r$. We will revise this notation to make it less confusing.
>
> > You use [1] as the ref for MSSA -- how is it different from earlier MSSAs that you describe in appendix?
>
> The main difference between the earlier mSSA and our version is in the time series matrix representation. Original SSA (and mSSA literature) utilizes the Hankel matrix representation - the version of mSSA that we use utilizes Page matrix instead (Definition 2.1).  This variant was proposed by [1] for tasks of forecasting and imputation, and we extend its use to CPD. This is explained briefly in our paper in lines 508-512 in the appendix, and we will bring that discussion up to the main paper in our revision.  Note that while this is a seemingly minor difference, it plays a big role in the theoretical analysis as well as offers improved empirical results (see the difference between Microsoft SSA and our algorithm in univariate time series). Indeed, this representation enables our novel analysis of the ARL metrics.
>
> > Property 2.2. The language makes it confusing as to whether it's an assumption or implication.
>
> It is an assumption. We will revise the wording of the property to make it more clear.
>
> > For finite sums of harmonics, low-degree polys, and exp-functions Page matrix is low-rank. Comment on rank vs. num terms? In general for your data-sets in the experimental section, what rank did you end up choosing (for default / best params)?
>
> The rank depends on the number of mixtures (each mixture is a product of a single harmonic, exponential, and polynomial) and the maximum degree of the polynomials. Specifically, the rank of $K$ mixtures with a maximum polynomial degree of $d$ is $K(1+d) (2+d)$. Refer to proposition 2.1 in [1] for more details.
>
> For choosing the rank, we follow the spectrum energy thresholding rule described in Equation (57) (Page 31 in the Appendix) for both best and default parameters (default choice being 95% of the spectrum energy).
>
>
> We again thank the reviewer for their comments and constructive feedback that has allowed us to run additional insightful experiments. We hope the reviewer takes our clarifications into account in their revised scores.

---

> > ### Comment · Reviewer_xiRb · 2021-08-27
> > **thank you for the thorough and detailed response**
> >
> > I'd like to thank the authors for a thorough and detailed response.  This was helpful!

---

> > > ### Author Response · Authors · 2021-08-27
> > > **Response**
> > >
> > > We thank the reviewer again for their detailed review and feedback and we will be happy to answer any further questions about the work.

---

### Official Review · Reviewer_TMbJ · 2021-07-16

**Rating:** 6
**Confidence:** 4

**Summary:**

This paper studies the online change-point detection for multivariate time series and proposes a new cusum-like detection statistic based on singular spectrum analysis. Theoretical approximations for the ARL and EDD are provided. This paper also validates the performance of the proposed method using a variety of synthetic and real data sets.

**Limitations And Societal Impact:**

Yes

**Main Review:**

The main contribution of this paper is the proposed detection statistic for online change detection, together with theoretical analysis on two major performance metrics: the average run length when there is no change, and the average detection delay.
The basic idea in this paper is to first use historical data to estimate the nominal subspace of the Page matrix, then under the assumption that the change will affect the corresponding subspace, the change can be detected by monitoring the deviation of the sequential observations with respect to the nominal subspace.

The construction of the detection statistic is only of incremental contribution since the deviation of the observation with a referenced subspace has been used in previous literature for change detection, and the construction and properties of the Page matrix are also known. However, this paper also provides a complete theoretical analysis on ARL_0 and ARL_1 which matches with classical results for iid data, and these theoretical results seem to be relatively new.

This paper may still be improved in the following directions.
First, it would be interesting to list some key constants (those C's) in Theorem 4.2 and Theorem 4.1 -- then we may obtain a more clear relationship between ARL_1 and ARL_0 and compare with the theoretical lower bound (the optimal detection performance).
Second, in the numerical results, the author spent a lot of space comparing the F1 score for different algorithms. Since this is an online algorithm, maybe the detection delay (and the trade-off between ARL_0 and ARL_1 as shown in Figure 3) would be more interesting to present. For example, instead of showing the trade-off curves for different N values, it would be more interesting to compare the trade-off curves for different algorithms.

**Time Spent Reviewing:**

3

---

> ### Author Response · Authors · 2021-08-10
> **Response**
>
> We thank the reviewer for pointing out the novelty of the results presented in the paper and for providing valuable feedback on ways to improve our empirical experiments and the presentation of the theoretical results. Below we address the specific areas of improvement suggested by the reviewer.
>
> > First, it would be interesting to list some key constants (those C's) in Theorem 4.2 and Theorem 4.1 -- then we may obtain a more clear relationship between ARL_1 and ARL_0 and compare with the theoretical lower bound (the optimal detection performance).
>
> We refer the reviewer to the proofs of Theorems 4.1 and 4.1 (Appendix B.3 and B.4), in which we derive the exact expressions of these constants. Due to space constraints, we omitted these constants in the theorems' statements.
>
> > Second, in the numerical results, the author spent a lot of space comparing the F1 score for different algorithms. Since this is an online algorithm, maybe the detection delay (and the trade-off between ARL_0 and ARL_1 as shown in Figure 3) would be more interesting to present. For example, instead of showing the trade-off curves for different N values, it would be more interesting to compare the trade-off curves for different algorithms.
>
> We thank the reviewer for suggesting this interesting experiment. However, we note that there are hurdles to applying such an experiment. First, such a trade-off is only meaningful for online algorithms. Second, to evaluate this tradeoff, we need long intervals between change points to be able to capture exponential running lengths; this is not entirely possible in real-world data because the average interval between change points is small. Nonetheless, we agree that comparing this tradeoff between *online* algorithms in *synthetic* data is interesting, and we will perform these experiments in our revision.
>
> We again thank the reviewer for their comments, constructive feedback, and interesting suggestions.

---

### Official Review · Reviewer_iaBb · 2021-07-19

**Rating:** 6
**Confidence:** 4

**Summary:**

The article presents a change point detection (CPD) method for multivariate data. The proposed methodology operates by monitoring the space generated by segments of the data and then assesses how this space varies. A sound theoretical treatment and convincing experiments are reported as well.


**Limitations And Societal Impact:**

yes

**Main Review:**

The paper is dense in notation and definitions that are needed to present the contribution, however, the authors present them in a very clear manner. Among the positive points of this work I would like to mention the fact that the proposed method is nonparametric, meaning that no model is imposed to the data, and also online, meaning that CPD is performed as observations arrive by evaluating the CUSUM statistic. The proposed method is multivariate and exploits across-channel relationships, a necessary feature for real-world applications. The ideas presented in the paper are also supported on a theoretical analysis, a discussion on the optimal hyperparameters and two sets of experiments.

I have the following criticisms

- The proposed method is for discrete-time observations, in the spirit of classical digital-signal processing techniques. This is undesirable for real-world scenarios for which the observations are asynchronous or there are (regions of) missing data. I know this paper targets another particular setting, but I suggest the authors refer to that case.

- The noise model, and the randomness, associated to the approach may be overly simple. Recently, there has been Gaussian process based CPD methods, which despite having several drawbacks are very capable in modelling complex trajectories. This proposed method, however, only considers a fixed structure with iid noise, which might be unable to detect changes in the form of, e.g., the distribution of the noise or the covariance of successive values of f_0(t).

- As the proposed method focuses on monitoring the distance of new points wrt the span of the previous data, how would the method behave for a nonstationary signal which changes very gradually and there is no changepoint per se but rather a continuous model evolution.


particular comments:

l111: denotes -> denote
l191: why is that choice for L recommended? Please elaborate
Table2:Summery -> Summary



**Time Spent Reviewing:**

2

---

> ### Author Response · Authors · 2021-08-10
> **Response**
>
> We thank the reviewer for the positive comment on the capabilities of the proposed algorithm and the comments and feedback that open doors for an interesting discussion about future work and improvements. Below we address each of these comments and questions.
>
> > The proposed method is for discrete-time observations, in the spirit of classical digital-signal processing techniques. This is undesirable for real-world scenarios for which the observations are asynchronous or there are (regions of) missing data. I know this paper targets another particular setting, but I suggest the authors refer to that case.
>
> We agree that change point detection in continuous (or asynchronous) time is an interesting research direction that is relevant in many real-world applications. However, we note that the majority of the literature on CPD considers the discrete time setting (including the methods we consider in our benchmark). We believe, as witnessed in the literature, that the discrete-time setting is effectively used to address many real-world change point detection problems like those discussed in [35] and [36] in our references, among many others.
>
> > The noise model, and the randomness, associated to the approach may be overly simple. Recently, there has been Gaussian process based CPD methods, which despite having several drawbacks are very capable in modelling complex trajectories. This proposed method, however, only considers a fixed structure with iid noise, which might be unable to detect changes in the form of, e.g., the distribution of the noise or the covariance of successive values of f_0(t).
>
> We would like to draw the reviewer’s attention to the fact that our model is actually capable of expressing complex time series trajectories. That is, even without the simple per-step noise model that we propose, we can model and detect changes in a rich class of time series dynamics. To explain this further,  traditionally, time series are modeled by three components: (a) stationarity, (b) periodicity, and (c) trend. Periodicity is modeled as harmonics and trend as polynomials; the mixture of both are instances of our model as discussed briefly in lines 106-110 (see more details in Proposition 2.1 in [1]).
> We agree that the inclusion of our model for stationarity is less obvious, however, by recalling the spectral representation of stationary processes (cf. see Property 4.1, Ch. 4 of “Time Series Analysis” by Shumway and Stoffer), which states that any sample-path of a stationary process can be decomposed into a sum of harmonics. Therefore, a finite sum of harmonics (which can be represented in our model) provides a good model representation for stationary processes, with the model becoming more expressive as the number of harmonics grows.
> In short, our spatio-temporal model representation captures *all* three types – stationarity, periodicity, and trend. Further, the additive noise measurement error in our model is an additional challenge that captures corruption present in real-world data.  We are grateful to the reviewer as this remark will help us explain the representational power of our model more clearly.
>
> > As the proposed method focuses on monitoring the distance of new points wrt the span of the previous data, how would the method behave for a nonstationary signal which changes very gradually and there is no changepoint per se but rather a continuous model evolution.
>
> * If by gradual changes the reviewer is referring to non-stationary processes, then our method is capable of handling such processes, e.g., low degree polynomials. This non-stationarity is not considered a change point as long as the underlying function that models the dynamics remains the same. We define change points in a concrete manner as the deviation from the nominal subspace spanned by the Page matrix of the time series.
> * Otherwise, if by gradual changes the reviewer refers to the case when the change point does not occur abruptly, but rather occurs gradually over time, then we refer in our discussion to the two implementations, “fixed base matrix” and “moving window”. The fixed base matrix implementation would be more sensitive to such gradual changes as the subspace is constructed at the beginning of the process. While in the moving window implementation, the subspace is updated with every time step making it harder to capture small changes that accumulate over time. This is for practical considerations;  theoretically, our bounds only address “abrupt” changes, and we believe addressing gradual changes is an interesting direction for future work.
>
> > 191: why is that choice for L recommended? Please elaborate
>
> With the choice of $L = \lfloor\sqrt{T_0\times\min(N,T_0)}\rfloor$,  we minimize the subspace estimation error ($\epsilon$). More concretely, the estimation error scales inversely with the square root of the minimum dimension of the Page matrix (refer to Proposition 4.1 and its proof in appendix B.2). Thus, to minimize the error, we construct a nearly square base matrix. We agree that this can be made clearer, and we will do so in our revision.
>
>
> We again thank the reviewer for their comments and for constructive feedback and suggestions. We hope the reviewer would take these clarifications into account in their revised scores.

---

> > ### Comment · Reviewer_iaBb · 2021-08-24
> > **rebuttal acknowledgement**
> >
> > I would like to thank the reviewers for their thorough responses. Based on the rebuttal and the rest of the reviews, I raise my score from 5 to 6.

---

> > > ### Author Response · Authors · 2021-08-27
> > > **Response**
> > >
> > > We thank the reviewer for considering our responses and for revising the score. We will be happy to address any further questions or concerns about the work.

---

### Official Review · Reviewer_e36y · 2021-07-29

**Rating:** 6
**Confidence:** 4

**Summary:**

This paper presents a multivariate singular spectrum analysis to perform change point detection. A spatio-temporal model is introduced to model the dynamics and CUSUM statistic was used to perform online change point detection. The experiment results showed the effectiveness of the proposed method.

**Ethical Concerns:**

No.

**Ethics Review Area:**

["I don’t know"]

**Limitations And Societal Impact:**

Yes.

**Main Review:**

Strengths:
* This paper is well organized.
* Change point detection is an interesting problem to investigate.
* Both theoretical and empirical justifications are provided for the proposed technique.

Weaknesses:
* It is not clear how the derived upper bounds are helpful for practical CPD.
* More discussions about the results are necessary.

Overall, the proposed multivariate singular spectrum analysis is innovative and technically sound. The following are a few questions:

It is not clear how the derived theoretical performance guarantee is related to performance improvement. Further explanations and discussions are necessary.

In Table 1, it is not clear why performance improvement is limited on Yahoo but very significant on Occupancy, Mixed, and Frequency.

Why is the variance of the proposed mSSA relatively large on several datasets in Table 2 and Table 3?

What’s the length of those time series datasets?


**Time Spent Reviewing:**

2 hours

---

> ### Author Response · Authors · 2021-08-10
> **Response**
>
> We thank the reviewer for highlighting the growing interest in the change point detection problem and the positive comments on the proposed mSSA algorithm. We also thank the reviewer for the helpful feedback on improving the clarity of our work in terms of theoretical results and discussion of empirical experiments. Below we address the specific questions and comments raised by the reviewer.
>
> > It is not clear how the derived theoretical performance guarantee is related to performance improvement. Further explanations and discussions are necessary.
>
> The average running length measures the number of time steps the algorithm takes before firing an alarm. If no change happens ($ARL_0$), a larger ARL is desired as it translates to a lower false-positive rate. When a change point occurs ($ARL_1$), a small ARL is desired as it corresponds to the detection delay. Our results theoretically characterize this tradeoff between $ARL_0$ and $ARL_1$.
> That being said, the reviewer raises a valid concern about using a different metric in the experiments. Although the ARL metric doesn't directly translate to F1-scores, we preferred using the F1-score as our metric due to two practical reasons: (1) it is more commonly used in the change point detection literature, which will make our results more accessible to the community; (2) we choose it over the detection delay metric (i.e., $ARL_1$) as it can be used for both online and offline methods, both of which are used in our experiments. We also note that the "truncation" of ARL as desired in practice, i.e. whether a change point was detected in some finite time or not, leads to a binary classification problem which naturally connects the analysis of ARL with F1-score with "very large" truncation threshold. We thank the reviewer for pointing this out, and we will make sure to clarify the reasons for this choice of metric in our revision.
>
> > In Table 1, it is not clear why performance improvement is limited on Yahoo but very significant on Occupancy, Mixed, and Frequency.
>
> We agree with the reviewer that an in-depth analysis of why the improvement is more significant in some datasets over others should be included in the paper. Based on our analysis, we believe that there are two main properties of the data that contribute to how well our proposed mSSA algorithm performs. The first one is to test whether the dataset follows our spatio-temporal model, for which Proposition 2.1 suggests a suitable data-driven diagnosis. Specifically, we verify that by testing whether or not the stacked Page matrix has a low effective rank (defined in  Equation (57)).  The second property is the distance and affinity between the pre-change and post-change subspaces. The distance measures the sine of the maximum angle between the subspaces and the affinity measures the cosine of the minimum angle between them. Since our devised detection score measures the projection of the lagged vectors onto the (estimated) pre-change subspace, these two quantities are very relevant to the algorithm's performance. We thank the reviewer for drawing our attention to this issue, and we will empirically investigate these two properties for our datasets in our revision.
>
> > Why is the variance of the proposed mSSA relatively large on several datasets in Table 2 and Table 3?
>
> As a clarification, we note that the mean and standard deviations we report in Table 3 are calculated across time series in the dataset (e.g., across the 99 time series in the Yahoo data). We do so to give a concise representation of the results. With that in mind,  we note that despite having a relatively higher variance (especially on Beedance and HASC datasets) than other methods, the ranges for mSSA performance are better/comparable to other methods. Below we report the best and worst-case performance of each algorithm on these datasets.
> Lastly, we note that our method's variance is comparable and sometimes lower than the other methods (e.g., Energy, Mean, Mixed, Frequency, and Yahoo datasets when using the best parameters set).
>
> | Method        | HASC (min-max F1 score) | Beedance (min-max F1 score) |
> |---------------|-------------------------|-----------------------------|
> | BinSeg        | 0.174 - 0.56              | 0.436 - 0.722                 |
> | Microsoft SSA | 0.154 - 0.375             | 0.508 - 0.700                 |
> | BOSPDMS       | 0.118 - 0.33              | 0.100 - 0.286                 |
> | KL-CPD        | 0.143 - 0.174             | 0.356 - 0.494                 |
> | mSSA MW       | 0.148 - 0.526             | 0.44 - 0.786                  |
>
> > What’s the length of those time series datasets?
>
> We list the length of the time series in each dataset below, and we will include this information in Table 2 in our revision.
> - Beedance: 6 time series each with observations between 608 and 1124.
> - HASC: 18 time series each with observations between 11738 and 12000.
> - Occupancy: 1 time series with 8143 observations
> - Yahoo: 99 time series each with 1680 observations
> - Energy, Mean, Frequency: 20 time series each with 5000 observations
> - Mixed: 5 time series each with 1000 observations
>
>
> We again thank the reviewer for their comments and constructive feedback.

---

> > ### Comment · Reviewer_e36y · 2021-08-31
> > **Thanks for the response**
> >
> > Thanks a lot for the response. Most of my previous concerns are resolved. I would suggest the authors incorporate these contents into the updated version to make it more clear and complete.

---

### Decision · Program_Chairs · 2021-09-27

**Decision:**

Accept (Poster)

**Comment:**


The paper studies sequential change-point detection using a combination multi-variate singular spectrum analysis (SSA) and CUSUM.  The paper presents a novel analysis of the online CPD setting with a dependent (non-i.i.d.) sequence and characterizing the tradeoff of detection delay vs. false-alarm. Extensive numerical results validate the performance of the algorithm.